# Solid Lipid Nanoparticles (SLNs) and Nanostructured Lipid Carriers (NLCs) as Food-Grade Nanovehicles for Hydrophobic Nutraceuticals or Bioactives

Chuan-He Tang [1,2,*] , Huan-Le Chen [1] and Jin-Ru Dong [3]

1   The Research Group of Food Colloids and Nanotechnology, School of Food Science and Engineering, South China University of Technology, Guangzhou 510641, China
2   Key Laboratory of Meat Processing of Sichuan, Chengdu University, Chengdu 610106, China
3   College of Food Science and Engineering, Ocean University of China, Qingdao 266100, China
*   Correspondence: chtang@scut.edu.cn

**Abstract:** Although solid lipid nanoparticles (SLNs) and nanostructured lipid carriers (NLCs) have been successfully used as drug delivery systems for about 30 years, the usage of these nanoparticles as food-grade nanovehicles for nutraceuticals or bioactive compounds has been, relatively speaking, scarcely investigated. With fast-increasing interest in the incorporation of a wide range of bioactives in food formulations, as well as health awareness of consumers, there has been a renewed urge for the development of food-compatible SLNs and/or NLCs as nanovehicles for improving water dispersibility, stability, bioavailability, and bioactivities of many lipophilic nutraceuticals or poorly soluble bioactives. In this review, the development of food-grade SLNs and NLCs, as well as their utilization as nanosized delivery systems for lipophilic or hydrophobic nutraceuticals, was comprehensively reviewed. First, the structural composition and preparation methods of food-grade SLNs and NLCs were simply summarized. Next, some key issues about the usage of such nanoparticles as oral nanovehicles, e.g., incorporation and release of bioactives, oxidative stability, lipid digestion and absorption, and intestinal transport, were critically discussed. Then, recent advances in the utilization of SLNs and NLCs as nanovehicles for encapsulation and delivery of different liposoluble or poorly soluble nutraceuticals or bioactives were comprehensively reviewed. The performance of such nanoparticles as nanovehicles for improving stability, bioavailability, and bioactivities of curcuminoids (and curcumin in particular) was also highlighted. Lastly, some strategies to improve the oral bioavailability and delivery of loaded nutraceuticals in such nanoparticles were presented. The review will be relevant, providing state-of-the-art knowledge about the development of food-grade lipid-based nanovehicles for improving the stability and bioavailability of many nutraceuticals.

**Keywords:** solid lipid nanoparticles (SLNs); nanostructured lipid carriers (NLCs); nanovehicle; nutraceuticals; stability; bioavailability

## 1. Introduction

Over the past several decades, great progress has been achieved in the development of functional foods enriched with bioactive compounds, or nutraceuticals, with potential benefits to human health. This came to pass, not only due to growing awareness about the health of customers, but also thanks to the development of innovative nanotechnology in the food science sector [1–3]. Quickly accumulating evidence has continued to support the supposition that the consumption of functional foods, or nutraceuticals, via normal diet is associated with improved health conditions, as well as reduced risks of multiple chronic diseases [4]. However, many nutraceuticals and bioactive compounds are poorly soluble and have low oral bioavailability, which greatly limits their applications in health-benefiting food formulations. It has been widely recognized that the usage of appropriate nano-delivery systems is one of the most effective strategies to improve the

solubility (even in aqueous phase), stability, and bioavailability of many poorly soluble nutraceuticals and bioactives, such as polyphenols, curcuminoids, carotenoids, phytosterols, and others [2,5–7]. Among all the reported nano-delivery systems for nutraceuticals, lipid-based nanocarriers seem to be the most appropriate delivery systems for hydrophobic or lipid-soluble nutraceuticals, due to their biocompatibility with lipid matrices [8,9]. The high specific surface area of lipid-based nanocarriers is favorable for lipid digestion, as well as absorption of the loaded bioactives from the intestinal tract.

Oil-in-water nanoemulsions, less than 200 nm in diameter, are among the most promising lipid-based nanocarriers, with a wide range of applications in food, pharmacy, and cosmetics [10]. Despite the enhanced physical stability due to the strong Brownian motion of nanosized droplets against gravity, the bioactive compounds encapsulated in nanosized lipid droplets usually undergo a faster release and more rapid oxidation degradation, since they are short in diffusion distance to the interface of droplets and large in surface area. In order to overcome this limitation, solid lipid nanoparticles (SLNs), which use solid (rather than liquid) lipids to limit the diffusion of bioactives in lipid matrices and improve oxidative stability, were designed and developed at the beginning of the 1990s as alternative carriers for emulsions, liposomes, and polymeric nanoparticles in the drug industry [11]. SLNs, as drug carriers, have been confirmed to exhibit many advantageous features—non-toxicity, biocompatibility, high stability and drug loading capacity, prolonged drug release, superior cellular uptake, high bioavailability, and easy scale up, for instance—over traditional colloidal delivery systems, e.g., liposomes and polymeric nanoparticles [12–17]. In addition to the drug formulation, SLNs exhibit great potential applications in a wide range of other fields, including cosmetic and dermatological preparations [11], as carriers for lipophilic bioactives in foods [8,18–20], or as delivery systems for antimicrobials [21].

Thanks to the successful development of SLNs in the pharmaceutical field over the past several decades, a number of reviews were available in the literature that addressed different aspects of SLNs, including their chemical composition (lipids, emulsifiers or co-emulsifiers, and water) [13,19], fabrication techniques, production methods [11,12,14,15,22], stability and related properties [14,19,23], digestion and absorption [16,24,25], drug vs. bioactive release [14,23], administration routes and in vivo fate [22], and bioactivities of encapsulated bioactives (e.g., essential oils and flavonoids, [26]). In general, SLNs possess some common physicochemical and structural characteristics, e.g., particle size and morphology, stability, and release behavior, depending on their chemical composition and production processes, or whether some surface modifications have been performed. When lipophilic bioactives are introduced in the lipid phase, the encapsulation efficiency and loading capacity (LC) of these bioactives in SLNs are also affected by their solubility in the lipids and interactions between bioactives and lipid compounds.

Despite these versatile advantages, SLNs also exhibit a few disadvantages or limitations, e.g., low LC of drugs or bioactives, and drug expulsion due to the occurrence of polymorphic transition upon storage [12,19]. To overcome these limitations, a novel kind of lipid nanoparticle (referred to as nanostructural lipid carriers (NLCs)) was proposed and developed nearly two decades ago. Instead of using solid lipids as the core in SLNs, a blend of solid and liquid (unsaturated fatty acids) lipids are applied to formulate NLCs that remain in solid state. The occurrence of imperfect crystallization in the lipid core of NLCs (as illustrated in Figure 1) allows a larger space between fatty acid chains, to facilitate the incorporation of drugs or bioactives, and to alleviate expulsion during polymorphic transitions [19]. Thus, NLCs, as delivery systems for bioactives, exhibit several advantages over SLNs, such as enhanced LC and encapsulation efficiency, better physical stability (e.g., minimized bioactive expulsion upon storage), controlled release, and improved bioavailability of bioactives [27,28].

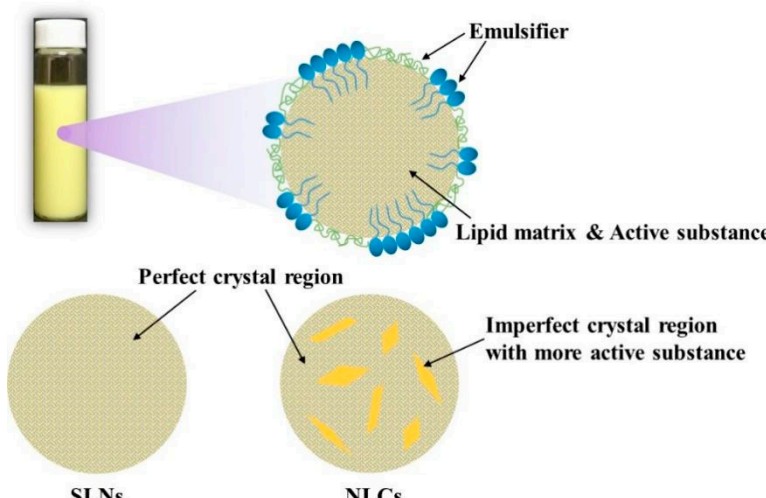

**Figure 1.** Schematic representation of the composition structures of lipid nanoparticles: SLNs and NLCs. (Adapted from [19]).

This article aimed to provide a systematic review of the usage of food-grade SLNs and NLCs as nanovehicles for hydrophobic (poorly water-soluble) or lipophilic (oil-soluble) nutraceuticals (and bioactive compounds). First, the structural compositions (lipids, emulsifiers, or co-emulsifiers) and preparation methods of such lipid-based nanoparticles were summarized. Next, some key issues on the usage of such nanoparticles as oral nanovehicles for hydrophobic bioactives, including incorporation and release of bioactives, oxidative stability, lipid digestion and bioactive solubilization, bioactive absorption, and intestinal transport, were critically discussed. Then, the utilization of SLNs and NLCs as nanovehicles for encapsulation and delivery of liposoluble or poorly soluble nutraceuticals and bioactives (e.g., carotenoids, fat-soluble vitamins, omega-3 polyunsaturated fatty acids, essential oils, curcuminoids, polyphenols, phytosterols, or phytostanols) was comprehensively reviewed. The encapsulation performance, stability, bioaccessibility/bioavailability, and bioactivities of curcuminoids (and curcumin in particular) in such lipid nanoparticles were highlighted. Lastly, strategies to improve the performance of such nanoparticles as nanovehicles for nutraceuticals were proposed, and future research trends and potential applications in food formulations were discussed.

## 2. A General Summary of Food Biocompatible SLNs and NLCs: Composition and Preparation Methods

Although most of the components frequently used to form SLNs in drug delivery are generally recognized as safe (GRAS), some GRAS-certified materials are not preferable for use in food formulations. On the other hand, many synthetic surfactants widely used in the fabrication of SLNs may have safety issues upon chronic ingestion. Thus, a need has arisen in the field: to develop food biocompatible SLNs by completely or partially substituting synthetic surfactants with natural emulsifiers, e.g., polysaccharides, proteins, and their mixtures.

### 2.1. Structural Composition

SLNs and NLCs are composed of lipid matrix (solid and/or liquid lipids), emulsifier, co-emulsifier, water and, if any included, bioactive compounds (or drugs). The lipid matrix can be a single solid lipid with a highly crystallized state or a combination of different lipid classes with different physical properties, e.g., melting temperatures. The SLNs or NLCs are generally stabilized by a surfactant layer, which may consist of a single surfactant, or be composed of a mixture of surfactants (emulsifiers and co-emulsifiers). The mobility of incorporated bioactive compounds in these nanoparticles is largely determined by the physical state of the lipid matrix [19,20]. The selection of lipid matrices for the

formulation of SLNs or NLCs as delivery systems for bioactive compounds should consider the following aspects:

(i).   the solubility of bioactive compounds in the lipid phase, and the efficiency of their incorporation;

(ii).  oxidative stability of the lipid phase, and storage stability of crystallized lipids in the nanoparticles;

(iii). use of food biocompatible lipid components, from the viewpoint of acceptable toxicological evaluation.

### 2.1.1. Lipids

The composition of the lipid matrix (as the core) is a crucial parameter when determining the properties, stability, and structure of SLNs and/or NLCs, especially the type of crystals that are formed upon cooling of the system. The solid lipid components used for SLNs and NLCs, which appear as solid state at ambient and body temperature, mainly consist of triacylglycerols (TAGs), fatty acids, monoacylglycerols, diacylglycerols, steroids, and waxes [19,22]. TAGs are composed of three fatty acids bound to a glycerol molecule (Figure 2, top). These fatty acids can be saturated or unsaturated, e.g., palmitic (C16:0) and stearic (C18:0) acids are the main saturated fatty acids, while oleic (C18:1) and linoleic (C18:2) acids are the representative unsaturated fatty acids in vegetable oils. Saturated fatty acids are stable against oxidation and possess a higher melting temperature than unsaturated fatty acids with the same chain length. The TAGs containing saturated fatty acids also usually exhibit higher melting temperatures than those with unsaturated fatty acids, due to better packing of the fatty acid chains.

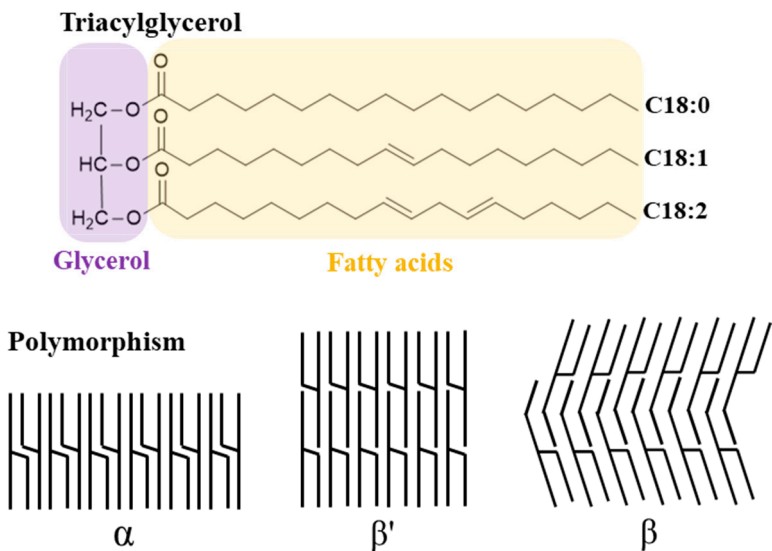

**Figure 2.** Schematic representation of triacylglycerol molecules and lipid polymorphism with three common types of subcellular cells, referring to the polymorphs $\alpha$, $\beta'$, and $\beta$.

Different solid lipids, as the core of SLNs or NLCs, exhibit different complex crystallization behaviors. During the cooling-induced solidification process, the crystallization of lipids takes places, and the polymorphic transitions of crystallized lipids further change during storage. To illustrate this, TAGs can be applied as an example. For TAGs, there are three main polymorphs ($\alpha$, $\beta'$, and $\beta$) with different lipid molecular packing (Figure 2, bottom). The $\alpha$-form is metastable with a hexagonal chain stacking, which tends to rapidly transform to $\beta'$- or $\beta$-forms with better packing and greater stability. Among all the forms of TAGs, the $\beta$-form is the most stable. Generally, a molten lipid (or lipid mixture) initially crystallizes as the $\alpha$-form during the solidification process, and the $\alpha$-form gradually and eventually transforms into the $\beta$-form via the $\beta'$-form during storage. The polymor-

phic transformation is an irreversible process, and is dependent on the cooling or storage temperature, time, and the degree of fatty acid heterogeneity of TAGs [19].

Stearic acid is a long-chain fatty acid, as a major component in many natural oils and fats, with high food compatibility and low toxicity. It has a high melting temperature of approximately 70 °C. This fatty acid and the corresponding TAGs (tristearin), or lipid mixtures rich in these components, have been widely utilized as lipid matrices to prepare SLNs [19]. Natural waxes, e.g., beeswax, can also be used as lipid matrices for food-compatible SLNs. They are commercially available, and have been approved as GRAS for food applications [19].

Regarding NLCs, liquid lipids with low melting temperatures are frequently mixed with solid lipids to form an imperfect crystalline structure, thus facilitating the drug or bioactive incorporation in the lipid matrix. The medium chain triacylglycerols (MCT), oleic acid and triolein are the most frequently applied lipid components for preparation of NLCs [19]. Oleic acid (C18:1), with a crystallization temperature of 4 °C, is a major component of most edible plant oils and fats. It possesses many health effects and is much more stable against oxidation than linoleic and linolenic acids. Therefore, this fatty acid is widely used as a liquid lipid to formulate NLCs. Aside from purified oleic acid, a wide range of edible oils, such as sunflower oil, flaxseed oil, corn oil and soybean oil, can also be used for NLC formulations [19]. The incorporation of liquid lipids in the lipid matrix of an NLC can alter its crystallinity, improve the solubility of encapsulated bioactives, and avoid the expulsion of loading components.

### 2.1.2. Emulsifiers and Co-Emulsifiers

Choosing an appropriate emulsifier (or surfactant) or emulsifier combination, as well as an appropriate emulsifier concentration, is of crucial importance for the production of SLNs [10]. At high temperatures above the melting temperature of the lipid matrix, the role of emulsifiers is to coat and stabilize the interface between molten lipid droplets and aqueous phase, thus facilitating the formation and stabilization of nanosized emulsions, especially those formed by high-temperature-homogenization (HPH) techniques. Except for lecithins, which are phospholipids, most emulsifiers (surfactants) are derived from mono- and diacylglycerols or from alcohols. Food grade emulsifiers mainly include mono- and diacylglycerols, phosphated mono- and diacylglycerols, propylene glycol esters, sorbitan esters, sucrose esters, polyglycerol esters, and lactate esters [19]. Many non-ionic emulsifiers, e.g., Tweens 20, 60 and 80, have been widely applied in stabilizing nanoemulsions via high or low energy methods, thanks to their high interfacial activities. Soy or egg lecithins (as representative zwitterionic emulsifiers) are also widely used in food formulations. For better formation and stability of nanoemulsions, a combination of nonionic and ionic emulsifiers may be used, since the presence of a certain concentration of ionic emulsifiers can produce a repulsive interaction between droplets, thus favoring emulsion stability. Natural lecithins are inefficient when used as the sole emulsifier to stabilize nanoemulsions; however, they might be efficient in combination with co-emulsifiers [10]. Co-emulsifiers, e.g., alcohols and short chain organic acids, can be used to improve the emulsification efficiency of some emulsifiers through increased flexibility of the interface of the emulsified systems.

The crystallization and oxidation of lipids in the nanoemulsions (or their SLNs or NLCs) stabilized by emulsifiers (and/or co-emulsifiers) may be affected by the type and concentration of emulsifiers (and/or co-emulsifiers). At higher concentrations of emulsifiers, nanoemulsions with smaller sizes are favorably formed. As a result, the interactions between lipid molecules and different functional groups of emulsifiers increase, thus modifying the crystallization kinetics and crystal polymorphs of lipids [20]. For instance, the use of high-melting-temperature lecithins as emulsifiers delayed the polymorphic transition from the metastable $\alpha$-form to the more stable $\beta$-form in SLNs with tristearin as the core [29] and inhibited the oxidation of omega 3 ($\omega$-3) fatty acids—which were used as the liquid lipid in NLCs that were mainly composed of tristearin [30].

Considering that many synthetic emulsifiers used in the production of SLNs (or NLCs) have potential health risks, e.g., cytotoxic effects [31], a need has arisen, in recent years, to develop more food biocompatible SLNs and NLCs using natural polymers (e.g., proteins, polysaccharides, or their complexes) to partially or completely substitute synthetic emulsifiers. Dr. Yangchao Luo's research group from the University of Connecticut made some interesting observations about the strategy and effectiveness of fabricating oral SLNs and/or NLCs, with Compritol ATO 888 (glyceryl behenate) as the lipid phase, using proteins and/or polysaccharides as emulsifiers or stabilizers to partially or completely substitute small-molecular-weight emulsifiers [32–35]. For example, they reported a novel strategy to prepare small and homogeneous SLNs using a combination of sodium caseinate and Tween 80 as the emulsifier and pectin as the coating polysaccharide. They demonstrated that Tween 80 (0.15%, *w/v*), together with caseinate (0.15%, *w/v*), was effective, utilizing it to obtain stable SLNs with a high loading capacity for curcumin [32]. The polysaccharide-coated SLNs were confirmed to exhibit exceptional gastrointestinal (especially stomach) stability and controlled release behavior (of encapsulated curcumin) [32]. They successfully fabricated what were essentially all-natural SLNs without the use of any synthetic surfactants via a combined solvent-diffusing and hot homogenization method [33]. The SLNs were formed through a layer-by-layer process, with soy lecithin as the natural emulsifier (first layer), followed by the sequential coating of sodium caseinate (second layer) and pectin (third layer). Once they performed a nano spray-drying, these SLNs could be transformed into redispersible ultra-fine powders [33]. Aside from its use in SLNs, this nano spray-drying technology could be applicable for the preparation of NLC powders [34] and SLNs as effective oral delivery systems for curcumin [35].

### 2.1.3. Preparation Methods

All procedures for the preparation of nanoemulsions are potentially suitable for SLNs and NLCs, provided that all the procedures are performed at a temperature above the melting temperature of the lipid matrix of the system. Such preparation procedures include high pressure homogenization, solvent emulsification–evaporation, solvent emulsification–diffusion, solvent displacement, ultrasonication, micro-emulsification, spontaneous emulsification, and phase inversion [10]. Many procedures involve the use of organic solvents, and are thus inappropriate to be used for the fabrication of food-grade SLNs or NLCs. Furthermore, preparation procedures that require the use of high concentrations of synthetic emulsifiers and/or co-emulsifiers are not compatible with the food industry. On the other hand, scale-up (like scalability at the industrial level) is also a parameter used to determine appropriate preparation methods for SLNs and NLCs.

High pressure homogenization (HPH) and microfluidization have widely been considered the most effective techniques for fabricating SLNs and NLCs, since related devices with different capacities are commercially available. As such, they have been widely applied in the food industry [18,19,22,28]. HPH has been used to produce nanoemulsions for parenteral nutrition; its scaling up, in most cases, has not presented any problems [22]. There are two homogenization approaches (hot and cold) used in the production of SLNs (Figure 3a). In both processes, the melting of lipids at high temperatures, followed by the incorporation of bioactives in molten lipids, is a prerequisite for production of SLNs. The hot HPH process is suitable for the loading of heat-stable bioactives in SLNs; this is generally carried out at temperatures above the melting temperature of the lipid matrix. At first, the bioactive-loaded lipid melt is added into a hot aqueous solution containing emulsifiers and/or co-emulsifiers. The mixture is then subjected to a shearing homogenization to form a coarse emulsion (at the same temperature as above). The coarse emulsion is further homogenized at high pressures, resulting in the formation of nanoemulsions. If necessary, the HPH treatment can be repeated several times to ensure that the resultant emulsions are at the nanosized scale. Lastly, SLNs are formed by cooling hot nanoemulsions to room temperature or below. As for the cold HPH process, it is more suitable for the encapsulation of heat-labile bioactive compounds [19]. In this case, the molten lipid phase, with bioactives

incorporated, is rapidly cooled with liquid nitrogen or dry ice, which ensures that all the loaded bioactives can be evenly distributed in the solid lipid matrix. Then, the solid matrix is mechanically ground into microparticles, with sizes in the range of 50–100 μm. After that, the microparticles are dispersed in a cold emulsifier solution to form a suspension. The suspension is further treated by HPH at room temperature or below, to form a dispersion containing SLNs.

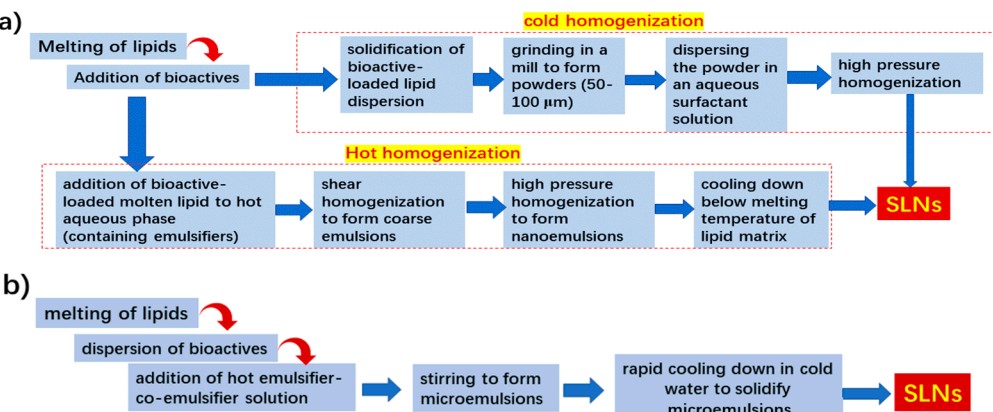

**Figure 3.** (**a**) The the hot and cold homogenization procedures used to obtain bioactive-loaded solid lipid nanoparticles (SLNs). (**b**) The melt micro-emulsification process used to obtain bioactive-loaded SLNs.

Aside from HPH, SLNs and NLCs can also be obtained using other high-energy-input emulsification techniques, e.g., microfluidization and ultrasonication [18,19,28]. In these cases, a pre-emulsification through a shearing process at temperatures above the melting temperature of lipid matrix is still a prerequisite for successful production of SLNs or NLCs. In contrast, the lipid nanoparticles formed by microfluidization would be much more homogeneous and finer than those made by ultrasonication. More importantly, industrial-scale microfluidizers are commercially available; thus, microfluidization exhibits great potential for use in the preparation of SLNs or NLCs.

In addition, SLNs or NLCs can be fabricated by many low-energy emulsification processes that do not require significant energy to reduce the emulsion size [28]. A microemulsion-based process for fabricating SLNs, first proposed and described by Gasco [36], is one such low-energy procedure, and is the most frequently applied, according to the literature [37–40]. Microemulsions are formed through the spontaneous self-assembly of the surfactant molecules in a solvent, giving rise to assembled architectures, such as micelles, bilamellar phases, and reverse micelles. These surfactant architectures, usually at sizes ranging from several to several tens of nanometers, can solubilize significant amounts of lipids or lipophilic components [20]. Figure 3b illustrates the general procedure for fabricating bioactive-loaded SLNs through a microemulsion-based process. The initial process of this procedure is identical to HPH and involves the melting of lipids (at 65–70 °C) and the incorporation of bioactives in molten lipids. The melt containing the bioactives is added to the hot solution containing emulsifiers (e.g., lecithin, polysorbate 20 or 60) and/or co-emulsifiers (e.g., bile acids and butanol). After that, the hot, optically-transparent mixture is stirred to induce the formation of microemulsions with the bioactive-lipid as the core (Figure 3b). The hot microemulsion is diluted in cool water (at 2–3 °C) under stirring to solidify the lipid matrix in the system. The dilution process is critically important for the successful fabrication of homogeneous SLNs, which is determined by the composition of microemulsion. The volume ratio of hot microemulsion to cold water is usually in the range of 1:25 to 1:50 [22]. In a more recent work, Shah and others [41] proposed a one-pot microwave-assisted microemulsion process to fabricate SLNs as nanocarriers for drugs. In this process, stearic acid, Tween 20, and water were heated at 80 °C for 10 min in a

microwave reactor to initiate the synthesis of the microemulsion. When preparing the drug-loaded SLNs, the drugs were added to the reactor tube during synthesis.

## 3. Some Key Issues on the Use of SLNs and NLCs as Oral Nanovehicles for Nutraceuticals or Bioactives

The development of food-grade nanovehicles, as delivery systems for improving water solubility (or dispersibility) and bioavailability of poorly soluble bioactive compounds, has become an intense research focus during the past decade, in both the food science and colloid fields. In general, an ideal food-grade nanovehicle, as a bioactive or nutraceutical, should meet some basic requirements, as follows: (1) high capacity to load significant amounts of bioactives; (2) site-targeted delivery; (3) high stability under stressed environmental conditions (e.g., temperature, oxygen, and light) and during consumption (e.g., pH, ionic strength and enzymes); (4) easy to prepare and scale-up with low cost; (5) high safety and no use of organic solvents; (6) commercial availability of raw materials in abundance; (7) good suitability to industrial drying processes (e.g., spray-drying or freeze-drying), ensuring the versatility of applications in food formulations [18].

The selection of a lipid matrix is a determining factor when developing an appropriate delivery system, based on SLNs or NLCs, for a specific bioactive. The solubility of bioactive in the lipid matrix and the interaction between the molten lipid and bioactive are two important parameters that must be considered when choosing the lipid matrix of SLNs. Both the encapsulation performance and loading capacity of bioactives in these nanovehicles and the stability of the encapsulated bioactives are affected by the selection of lipids. On the other hand, the selection of the lipid matrix is also, to a certain extent, determined by the nature of bioactives, e.g., lipophilicity, charge, molecular weight, structure, and functional groups.

### 3.1. Incorporation and Release of Bioactives

Although SLNs (and NLCs), in their early history, were developed and utilized as nanosized delivery systems in the pharmaceutical field, there has been increasing interest in their application for the encapsulation and delivery of bioactives and nutraceuticals in the food industry over the past decade. To date, a great number of bioactive compounds, with a variety of physicochemical characteristics (e.g., lipophilicity) and structures, have been successfully incorporated in SLNs and NLCs (Tables 1 and 2). These bioactive compounds can be divided into three categories: (i) lipophilic bioactives, e.g., carotenoids ($\beta$-carotene, lutein, lycopene, astaxanthin, bixin), tocopherols and vitamin A, $\omega$-3 fatty acids or fish oil, essential oils (citral and eugenol); (ii) poorly soluble bioactives, e.g., phytosterols (or its purified components), curcuminoids (curcumin), resveratrol and quercetin; (iii) water-soluble bioactives, e.g., nisin, tea polyphenols [42,43].

The first important issue regarding the incorporation of bioactives in SLNs and NLCs is encapsulation performance and loading capacity (LC). The LC is generally expressed in percentage of loaded bioactive related to the lipid phase (matrix lipid + bioactive). In a drug case, Müller et al. [11] summarized that the LC of drug in the lipid of SLN was determined by solubility of drug in molten lipid, miscibility of drug melt and lipid melt, chemical and physical structure of solid lipid matrix, and polymorphic state of lipid matrix. Thanks to their lipophilic nature, many lipophilic bioactive compounds were expected to be loaded in the lipid matrix with a high LC. In contrast, the LCs of poorly soluble or water-soluble bioactives were relatively low in these nanovehicles. On the other hand, it is worth mentioning that the solubility of bioactives in lipid melt is reduced when the molten lipid is cooled; furthermore, the crystallization of lipids occurs upon cooling, which may further decrease the retention of the bioactives in the lipid matrix. It has been well recognized that the addition of solubilizers (e.g., emulsifiers) and the presence of mono- and diglycerides in the lipid matrix are two effective strategies to promote bioactive/drug solubilization or increase the LC [11]. The improved incorporation of bioactives in SLNs

and NLCs in this case was clearly associated with the inhibited lipid crystallization, and delayed polymorphic change of lipids (from unstable α-form to stable β-form).

There are basically three representative models to describe the incorporation of bioactive ingredients into SLNs: (i) homogenous matrix model (or solid solution model); (ii) bioactive-enriched shell model; (iii) bioactive-enriched core model [11,23]. The localization of bioactives in SLNs depends on the composition and nature (e.g., melting temperature) of solid lipid matrix, the solubility and miscibility of bioactives in the lipid matrix, and the preparation process (e.g., hot vs. cold homogenization). A homogenous matrix with molecularly dispersed bioactive (Figure 4a, left) is frequently observed when the cold homogenization method is applied, or when very lipophilic bioactives with high crystallization temperatures are incorporated in SLNs with the hot homogenization method. For example, β-carotene can be incorporated homogenously into a matrix of tristearin [23]. If the polymorphic transition of the lipids in the lipid matrix is inhibited, e.g., by addition of a stabilizing agent (e.g., high-melting lecithins), the incorporated bioactive can be evenly distributed in the α-form structure of the lipid matrix, surrounded by a layer of the stabilizing agent [44].

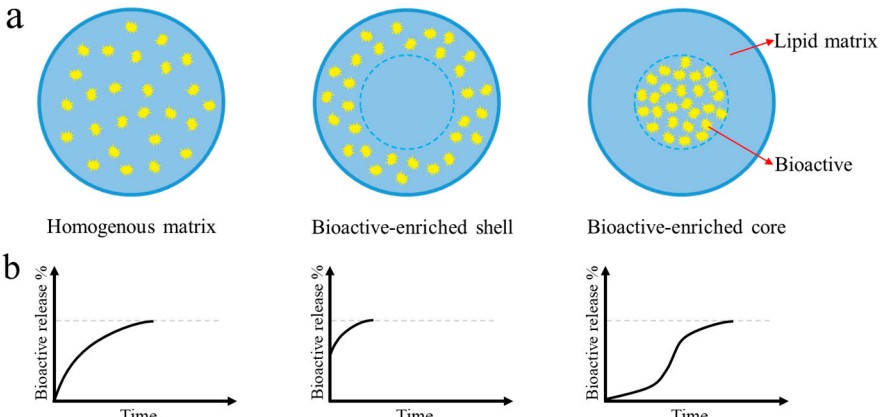

**Figure 4.** (**a**) Three models for the incorporation of bioactives into SLNs: homogenous matrix model (left), core-shell models with bioactive-enriched shell (middle), and bioactive-enriched core (right). (**b**) The release kinetics of bioactives from the SLNs with the corresponding three models (above).

In the hot homogenization process, the preparation of SLNs is frequently performed at least 10 °C above the melting temperature of the solid lipid. In general, a bioactive has greater solubility at higher temperatures. This means that the solubility of a bioactive in the lipid matrix decreases during the cooling process (to solidify). Once the aqueous phase becomes oversaturated with the bioactive as a result of cooling, the lipophilic bioactive tends to migrate to the dispersed phase (lipid). Here, it should be noted that, during the solidification process, both the bioactive and the molten lipid may crystallize, and phase separate. If the molten lipid of hot nanoemulsions first crystallizes to form a bioactive-free solid lipid nanoparticle, a core-shell SLN with a bioactive-enriched shell (Figure 4a, middle) is obtained, since the concentration of the bioactive in the remaining aqueous phase continuously increases. On the other hand, a core-shell SLN with bioactive-enriched core (Figure 4a, right) is formed when the bioactive precipitates prior to the crystallization of the molten lipids. In addition to the temperature-related effect, the incorporation of bioactives into SLNs is also affected by the emulsifier concentration in the aqueous phase; higher the emulsifier concentration, the greater the saturation solubility of the bioactive in the aqueous phase. In this case, a core-shell SLN with bioactive-enriched shell may be more favorably formed at higher emulsifier concentrations.

Regarding the location of a bioactive in NLCs, it is generally recognized that the location of the bioactive is closely related to the liquid lipid ratio (relative to solid lipid) and the relative miscibility of the bioactive in the liquid or solid lipids. Bioactives with a high lipophilicity, e.g., γ-oryzanol and β-carotene, are apt to retain in the liquid lipid or liquid

compartments in the lipid matrix of NLCs and, as a result, the expulsion of the bioactives from the lipid matrix during the cooling decreases [45,46]. Coenzyme $Q_{10}$ is an interesting lipophilic nutraceutical with a relatively low melting temperature of approximately 50 °C. In the hot HPH case, this bioactive is present in a liquid molten state, which can be well miscible with the molten solid lipid matrix. Previous works indicated that the localization of coenzyme $Q_{10}$ in the tripalmitin SLN was dependent on its concentration; at low concentrations of coenzyme $Q_{10}$ (below 10% of the lipid matrix), this bioactive was located inside the crystalline structure of the lipid matrix, while at high concentrations, it was located on the surface of the particles forming a supercooled lipid phase [47]. In fact, coenzyme $Q_{10}$-loaded SLNs at high concentrations of coenzyme $Q_{10}$ are considered to be a variety of NLCs, with the supercooled lipid phase of coenzyme $Q_{10}$ serving as the liquid lipid.

The release behavior of bioactives from SLNs (or NLCs) is highly determined by the localization of bioactives in the lipid matrix of SLNs (Figure 4). As mentioned above, the localization of bioactives in an SLN is dependent on the production temperature (hot vs. cold HPH) and emulsifier concentration (in the aqueous phase). The release behavior of a bioactive from an SLN or NLC is generally biphasic—that is, an initial burst release followed by a prolonged release [11]. The extent of burst release can be modulated by adjusting the production temperature and emulsifier concentration. Additionally, the surface modifications of the SLNs may modulate the bioactive or drug release behavior of SLNs or NLCs [23].

### 3.2. Oxidative Stability

In addition to the bioactive expulsion from the lipid matrix, the oxidation of lipids and oxidative degradation of encapsulated bioactives are another major stability problem for SLNs and NLCs in use as nanovehicles for bioactives. The selection of the lipid matrix is of crucial importance for the development of SLN- or NLC-based nanovehicles for heat- or UV-labile bioactives. In general, TAGs with long-chain fatty acids display better oxidative stability than those with short-chain fatty acids, and more unsaturated TAGs are less stable against oxidation than less unsaturated TAGs [19]. Using the hot HPH technique, the chemical stability of the lipid matrix for SLNs is unaffected by the production process, and some SLNs with TAGs as the solid lipid show excellent chemical stability at room temperature [48]. In an aqueous environment, the oxidation of lipids in SLNs or NLCs generally occurs at the oil–water interface, which may be initiated by metal ions, light, and oxygen in the system. For example, a previous work indicated that the dilution of the NLCs dispersion with water decreased the oxidative stability of their encapsulated $\beta$-carotene, largely as a result of increased dissolved oxygen [49]. Thus, the mobility of lipids in the lipid matrix (or physical state of the lipid matrix core) and the interface-stabilizing structure are two important parameters determining the lipid oxidation and degradation of encapsulated bioactives of SLNs and NLCs. For example, Helgason et al. [50] reported that the use of high-melting (HM) lecithin as the emulsifier remarkably inhibited the oxidation degradation of $\beta$-carotene encapsulated in the SLNs with tripalmitin as the solid lipid matrix. Meanwhile, in the case of liquid carrier lipids, no inhibition was observed. They attributed this high oxidative stability of SLNs to the role of the emulsifier (HM-lecithin) tails in the maintenance of the less polymorphic transition of lipid crystals. The same group further confirmed that the encapsulation of $\omega$-3 fatty acids in the NLCs stabilized with HM-lecithin remarkably inhibited the oxidation of these fatty acids, as a result of the formation of a solid surfactant interfacial layer [30].

In the case of conventional SLNs stabilized by Tween 80, the loaded $\beta$-carotene degraded faster than that in the liquid lipid nanoparticles after 8 days of storage [51]. The unexpected result was attributed to the occurrence of $\beta$-carotene expulsion in the lipid matrix due to the partial coalescence of lipid nanoparticles caused by the formation of fat crystals. Pan et al. [46] further indicated that $\beta$-carotene could be excluded from the lipid domain in both SLNs and NLCs, but the extent of exclusion decreased with an increase of

liquid lipid content of NLCs. However, when the liquid lipid content was 50% or above, the oxidative stability of the encapsulated $\beta$-carotene in NLCs was significantly better than that in emulsion, or the results were reversed [46].

### 3.3. Lipid Digestion and Bioactive Solubilization

A nanoparticle in food, after ingestion, experiences different stresses (e.g., pH, ionic strength, temperature, digestion fluids, mechanical mixing, and action of enzymes) of gastrointestinal (GI) processes, with respect to mouth, stomach, small intestine, and colon. Changes in the size and interfacial characteristics (e.g., $\zeta$-potential) of the nanoparticle may occur as it passes through different stages of the GI tract; these are highly dependent on the type of applied emulsifiers [24]. For lipid nanoparticles stabilized by Tween 20, as frequently observed in SLNs and NLCs, their particle size and $\zeta$-potential slightly change after the mouth and stomach stages of simulated digestion, but small-intestine digestion results in a significant increase in particle size (largely as a result of droplet flocculation and coalescence), and magnitude of $\zeta$-potential. This implies that most nanoparticles of SLNs or NLCs may exhibit a high degree og stability against gastric digestion, e.g., strong acid (pH 1–3), high ionic strength, and high enzyme activity (e.g., gastric lipases), as was confirmed in a simulated gastric and intestinal digestion work for trimyristin nanoparticles [52]. Upon entering the small intestine, the acidic gastric digest (of lipid nanoparticles) is mixed with the alkaline small intestinal fluids from gall bladder and pancreas, which contain bile acids (BA), phospholipids (PL), pancreatin lipase and co-lipase, and salts. As a result, the pH of the mixture increases up to around neutral. It should be mentioned that the secretion of endogenous biliary lipids (BS, PL, and cholesterol) from the gall bladder is significantly enhanced by the presence of exogenous lipids in the small intestine. In this regard, long-chain fatty acids seem to be much more effective in stimulating the secretion of bile by the gall bladder [25].

In the small intestine, pancreatic lipase, together with its co-factor co-lipase, is adsorbed at the interface of lipid nanoparticles (SLNs or NLCs), and catalyzes TG to release diglycerides, monoglycerides, and fatty acids [25]. If exogenous PL is present, e.g., as emulsifiers for lipid nanoparticles, the hydrolysis of PL by pancreatic phospholipase $A_2$ also occurs in the small intestine, yielding lysophophatidylcholine and fatty acids. Then, in the presence of endogenous BS and PL at enhanced concentrations, the products of lipid digestion (monoglycerides, fatty acids and lysophospholipids) are incorporated into a series of colloidal nanoarchitectures with a variety of structures, such as multilamellar and unilamellar vesicles, mixed micelles, and micelles [25]. The bioactives released from the lipid nanoparticles as a result of lipid hydrolysis can be favorably transferred into the interior of these nanoarchitectures, thus causing significantly improved solubility of these bioactives in the small intestine fluid. The detailed description for the process of lipid digestion (of SLNs or NLCs) and bioactive solubilization in the small intestine is illustrated in Figure 5.

The bioactive solubilization capacity of these colloidal architectures significantly increases upon addition of exogenous lipids (from formulation or foods) and is dependent on the nature of the digestion products (e.g., fatty acid chain length) and the structural characteristics of the colloidal architectures [25]. Thus, it can be reasonably expected that the bioaccessibility of bioactives encapsulated in the lipid nanoparticles (SLNs or NLCs), which is defined as the fraction of bioactives released from their lipid matrix and solubilized within the GI fluid that can be adsorbed [53], varies significantly with differences in the composition, concentration, and nature of lipids in their lipid matrix. For example, Noack et al. [52] observed that the transfer of curcuminoids in the SLNs into the simulated GI fluid was mainly triggered by the lipid degradation and not by the bioactive release. Yang and colleagues [54] further indicated that the release of encapsulated bioactives (e.g., curcumin) in the NLCs could be controlled by modulating the lipid type and composition, and by using a lipase inhibitor. A more recent work, interestingly, confirmed that the bioaccessibility of curcumin encapsulated in NLCs with glyceryl tristearate and medium-chain

triglyceride (MCT) as the solid and liquid lipids was closely correlated with the relative liquid lipid ratio in the total lipids, and a maximum value of bioaccessibility was obtained at the relative liquid ratio of 20% [55].

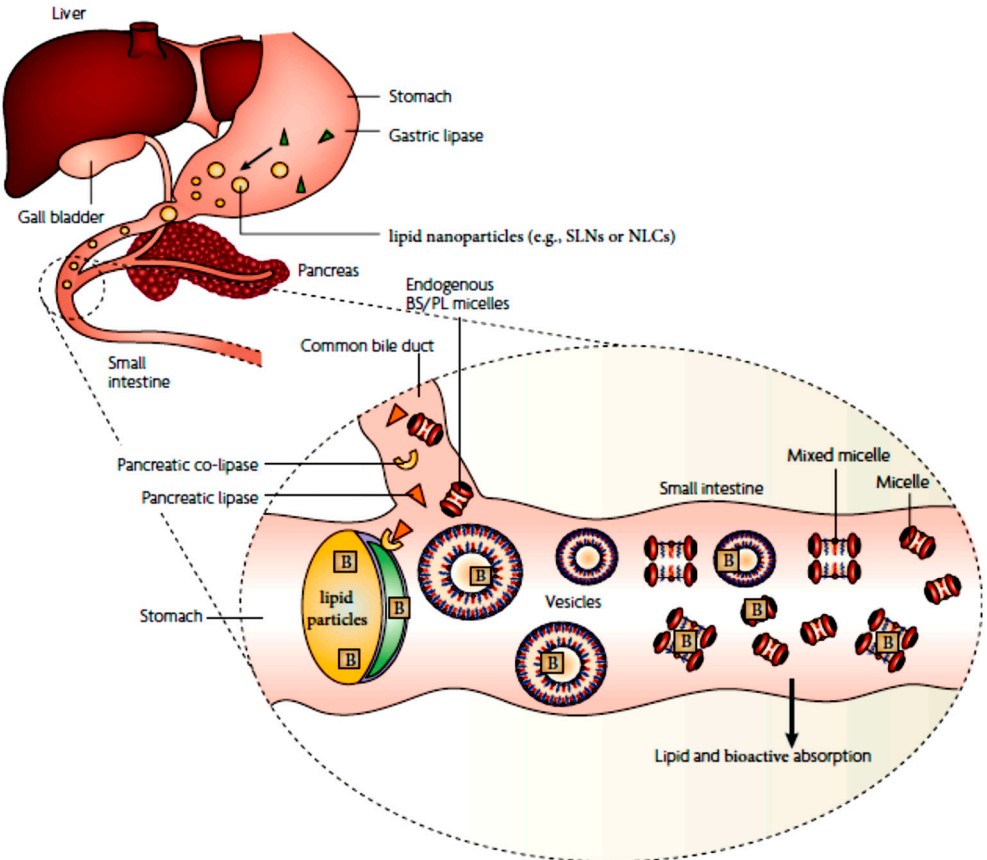

**Figure 5.** Schematic illustration for lipid digestion of lipid nanoparticles (SLNs and NLCs) and bioactives (B) solubilization in the small intestine. (Adapted from [25]).

*3.4. Bioactive Absorption and Intestinal Transport*

Before the adsorption though the epithelium cells of the GI tract, the bioactives encapsulated in the mixed micelles (formed after the digestion of lipid particles), or in the intact or partially digested lipid nanoparticles, may be penetrating into the mucous layer covering the surfaces of the enterocytes [25]. If the size of the colloidal structures containing bioactives is too large, they are unable to enter the mucus layer and will remain in the lumen of the GI tract. Here, there are two aspects of adsorption of bioactives that should be considered: (i) absorption of bioactives (solubilized in the mixed micelles); (ii) absorption of lipid nanoparticles [24]. If the lipids in SLNs and NLCs containing bioactives are completely digested by the enzymes during the digestion, before being transported to the adsorption site, only the absorption of bioactives is considered. On the other hand, if the lipid nanoparticles are composed of indigestible lipid components (e.g., essential oils, flavor oils, and fat replacers), or stabilized and coated by indigestible shell layers (e.g., polysaccharides), these particles may be absorbed directly into the systematic system via the portal vein and the intestinal lymph (paracellular or transcellular; see below)—with the caveat that they must penetrate through the mucous layer and be translocated through the epithelium cell layer [24,25,56]. In fact, mixed micelles and other colloidal structures naturally formed during lipid digestion, which usually have sizes < 100 nm, can also be considered a kind of lipid nanoparticles. The nanoparticles are taken up by passive diffusion, facilitated diffusion, and active transport across the enterocyte membrane of the GI tract [16].

The lipid composition, or formulation, of SLNs and NLCs plays a crucial role in the solubilization of their encapsulated bioactives in the mixed micelles (or other colloidal structures), and absorption across the epithelium cell layer. In the cases of certain drugs, drug solubilization or oral bioavailability in the presence of long-chain triglycerides is much more effective than that in the presence of medium-chain lipids [25]. This is largely associated with the facilitated formation of mixed micelles with more hydrophobic fatty acids, MAGs and DAGs incorporated, in the case of long-chain TGs. However, it should be noted that the solubility of poorly water-soluble bioactives (or drugs) is often low in long-chain triglyceride lipids, which may limit their loading capacity in these lipid systems. In general, the micellar solubilization of bioactives, together with free fatty acids, MAGs, or DAGs, greatly enhances the mass transport of these components across the mucous layer. Although there a currently-ongoing debate regarding whether or not micelles are absorbed intact across the unstirred layer in the small intestine, more evidence supports the supposition that free fatty acids (or other components in the mixed micelles) may dissociate in the unstirred layer before partitioning into the enterocyte [25]. These dissociated components are then absorbed across the apical membrane of the enterocyte via passive diffusion or carrier-mediated transport; alternatively, the undissociated micelles may be absorbed by a collisional transfer mechanism, or by a binding (to a transport protein on the apical membrane), followed by vesicular uptake [25].

Following uptake into the enterocytes, free fatty acids, MAGs, or DAGs can be re-synthesized to TGs in the smooth endoplasmic reticulum, and further assembled into TG-rich lipoproteins, e.g., chylomicrons and very low-density lipoproteins (VLDL). These lipoproteins are exocytosed from the enterocyte into the lamina propria, which preferentially access the lymphatics (across highly permeable lymphatic endothelium). After absorption, bioactives (or drugs) can also be transported to systematic circulation by either the portal vein or the intestinal lymphatic system. In most cases, the bioactives are absorbed through the portal vein, since the rate of fluid flow in portal blood is considerably higher than that of the intestinal lymph. However, for very lipophilic drugs (or bioactives), e.g., fat-soluble vitamins, retinoids, and lycopene, they tend to associate with the lipoproteins (above) and be transported in lymph within the apolar lipid core of lymph lipoproteins [25].

Lipid nanoparticles (including SLNs and NLCs) can be directly absorbed by the epithelium cells in the enterocyte via different transport mechanisms, as shown in many cases using in vitro Caco-2 models [57,58]. Roger et al. [58] indicated that the uptake of lipid nanoparticles (as a nanocarrier) by Caco-2 cells mainly occurred via active endocytic processes—and, more specifically, via clathrin-dependent and caveolae-dependent transport mechanisms. Due to this, the incorporation of paclitaxel (a hydrophobic drug) in lipid nanoparticles greatly improved its transport across the intestinal layer. In a more recent work, Neves et al. [57] compared the cellular uptake, internalization pathways and transcytosis routes through Caco-2 cell monolayers between SLNs and NLCs (with a similar size ~ 180 nm and surface charge −30 mV), and showed that NLCs had a higher cellular uptake and permeability across the barrier, while SLNs were more quickly absorbed by the cells than NLCs. Both nanoparticles were able to cross the intestinal barrier by transcellular or paracellular routes, but the transport mechanisms varied, depending on the type of lipid nanoparticles. The uptake of both SLNs and NLCs occurred mainly through a clathrin-mediated endocytosis mechanism; though in the case of SLNs, the caveolae-mediated endocytosis also played an important role in the uptake. Both the lipid nanoparticles were able to cross the intestinal barrier, predominantly through a transcellular route. NLCs were also confirmed to cross the intestinal barrier through a paracellular route, which agreed with the higher cellular uptake and permeability of NLCs [57].

## 4. Utilization of SLNs and NLCs as Nanovehicles for Encapsulation and Delivery of Hydrophobic Nutraceuticals or Bioactive Ingredients

Although the utilization of SLNs and NLCs as delivery systems for drugs has an approximately 40-year history, the application of such lipid-based delivery systems for nutraceuticals or bioactive components in the food industry has only about 20 years. Thanks to their lipophilic nature, lipid-based nanoparticles have advantages over other colloidal systems. They are able to perform as nanovehicles for liposoluble nutraceuticals, including carotenoids, vitamins A, D and tocopherols, $\omega$-3 fatty acids, and essential oils, since these nutraceuticals are completely compatible and miscible with the lipid matrix. Considering that many liposoluble nutraceuticals, e.g., carotenoids and $\omega$-3 fatty acids, are present in liquid oil form, or within liquid oils themselves, they can be used directly as the liquid lipid to fabricate NLCs with a high encapsulation efficiency (EE) [49], or as nanovehicles for co-loading another nutraceutical [59]. SLNs and NLCs have been recognized as the most innovative and promising nanovehicles to enhance the bioavailability of poorly adsorbed drugs [16]. Thus, the incorporation of liposoluble nutraceuticals in these lipid-based nanovehicles is expected to significantly improve their bioavailability, thus facilitating their applications in food formulations.

Aside from liposoluble nutraceuticals, many hydrophobic (but poorly soluble) nutraceuticals and bioactive components, e.g., curcumin or phytosterols, can also be encapsulated within SLNs or NLCs through different fabrication strategies. The nanoencapsulation of these poorly soluble bioactive components in lipid-based nanoparticles has been confirmed to be one of the most effective strategies for improving their availability [53]. Great progress has been made in the development of such lipid nanovehicles for curcumin over the past 10 years (see the below), due in large part, perhaps, to the following reasons: (i) curcumin is a golden bioactive, with many promising beneficial effects on human health [60,61]; (ii) lipid nanoparticles have been confirmed to act as effective nanovehicles for the delivery of drugs [17,62].

In this section, we have provided a critical review about the development of SLNs and/or NLCs as nanovehicles for many liposoluble nutraceuticals, with an emphasis on carotenoids with different structural and physicochemical properties, e.g., $\beta$-carotene, astaxanthin, lutein, lycopene, and bixin (Table 1). Then, advances in the utilization of SLNs and/or NLCs as potential nanovehicles for improving the water dispersibility, stability, and bioaccessibility (or bioavailability) of many poorly soluble bioactive components, e.g., curcumin and phytosterols, were comprehensively reviewed. Herein, issues regarding fabrication strategies for SLNs and NLCs, the stability and release behavior of encapsulated bioactives, as well as their in vivo health effects, were also fully discussed.

**Table 1.** A summary of some representative works addressing the development and/or utilization of SLNs and NLCs as nanovehicles for liposoluble nutraceuticals or bioactives.

| Liposoluble Nutraceuticals | Type of Lipid Nanoparticles [a] | Formulation Ingredients [b] | Processing Technique [c] | EE/DL [d] | Main Results | References |
|---|---|---|---|---|---|---|
| Carotenoids | | | | | | |
| $\beta$-carotene (10%; in sunflower oil) | NLCs | Propylene glycol monostearate; sunflower oil; Tween 80 | Hot HPH | -; 0.35% | The dispersions of $\beta$-carotene NLCs were stable at 4–8 °C after 30 weeks; the recrystallization of solid lipids occurred completely; the oxidative degradation of encapsulated $\beta$-carotene was enhanced by the dilution in water. | [49] |
| $\beta$-carotene | SLNs | Tripalmitin; lecithin or Tween 60 or 80 | Hot HPH | -; 0.1% | The crystallization of solid lipids was affected by the type of surfactants; the use of high melting surfactant (lecithin) in the SLNs provided a better protection against chemical degradation of encapsulated $\beta$-carotene. | [50] |
| $\beta$-carotene | NLCs | Palmitic acid; corn oil; Tween 20 | Solvent diffusion | 25–91%; - | The optimization of the formulation of $\beta$-carotene-NLCs was performed, with the aim to achieve the small particle size and high $\beta$-carotene retention; the NLCs were spherical in morphology, | [63] |
| $\beta$-carotene | NLCs | Milk fat; Tween 80 | Hot shear homogenization + phase-inversion-temperature | -;- | A kind of transparent $\beta$-carotene-NLCs dispersions were successfully fabricated; the NLCs showed stability against dilution, storage, and chemical degradation. | [64] |
| $\beta$-carotene | SLNs | Hydrogenated palm oil, cocoa butter; Tween 20 | Hot HPH | -; ~0.5% | The SLNs had better stability to droplet aggregation but lower $\beta$-carotene oxidation than liquid lipid nanoparticles; the impaired stability was attributed to the $\beta$-carotene exclusion from the crystallized lipids. | [51] |

| Liposoluble Nutraceuticals | Type of Lipid Nanoparticles [a] | Formulation Ingredients [b] | Processing Technique [c] | EE/DL [d] | Main Results | References |
|---|---|---|---|---|---|---|
| β-carotene | SLNs, NLCs | Eicosane, glyceryl trioctanoate (liquid); high-melting lecithin | Hot shear homogenization + ultrasonication | 16–100%; - | The exclusion of β-carotene occurred in the SLNs and NLCs, with the extent of exclusion decreasing with increasing the liquid lipid ratio; the oxidative stability of encapsulated β-carotene was much higher at 50–70% liquid lipid contents than that at 10–30% liquid lipid contents. | [46] |
| β-carotene | SLNs | Glyceryl stearate or hydrogenated palm oil (HPO); Tween 80 | Hot HPH | -;- | The solid lipid type affects the in vitro digestibility and β-carotene bioaccessiblity of SLNs; there was no close relationship between the digestibility and β-carotene bioaccessibiity. | [65] |
| lutein (20%; in corn oil) | NLCs | Glyceryl palmitostearate; Tween 80 and others | Hot ultrasonication | ~85%; 0.05–3.0% | The size of as-obtained NLCs depended on the treatment duration and lutein loading; the NLCs had a spherical morphology with an imperfect crystalline lattice structure, and showed a sustained-release delivery. | [66] |
| lutein (20%; in corn oil) | NLCs | cetyl palmitate, glyceryl tripalmitate, or wax; MCT; caprylyl/ capryl glycoside | Hot HPH | 89–100%; ~2.0% | The nanoencapsulation in the NLCs greatly protected the UV-induced degradation of lutein; the encapsulated lutein released in a biphasic manner. | [67] |
| lutein (20%; in corn oil) | NLCs | Glycerol stearate, wax; fish oil; Tween 80+lecithin/ poloxamer 407 | Hot shear homogenization | 50–88.5%; about 0.4–1.2% | The as-fabricated NLCs exhibited a high blocking effect against oxygen free radicals, and a good in vitro sustain release behavior. | [59] |

**Table 1.** *Cont.*

| Liposoluble Nutraceuticals | Type of Lipid Nanoparticles [a] | Formulation Ingredients [b] | Processing Technique [c] | EE/DL [d] | Main Results | References |
|---|---|---|---|---|---|---|
| lycopene | NLCs | orange wax; rice oil; sodium stearoyl glutamate | Hot HPH | 100%; 0.1–1.0 % | The fabricated lycopene-loaded NLCs showed a biphasic release profile, and exhibited an excellent colloidal stability upon storage of 120 days; the stability of lycopene was enhanced. | [68] |
| lycopene | SLNs | Glyceryl palmitostearate or glyceryl behenate; Tween 80 + Poloxamer 407 | Hot shear homogenization | 87–98%; 4.5–5.2 % | The size of SLNs was dependent on the type of applied solid lipids; the dispersions containing the lycopene-SLNs exhibited a good storage stability at 4 °C. | [69] |
| lycopene | NLCs, SLNs | GDS and GMS; MCT oil; lecithin + Tween 80 | Hot shear homogenization + ultrasonication | 65–79%; 4.54–5.52% | The EE of NLCs was significantly higher than SLNs; the nanoencapsulation improved the solubility of the bioactive in aqueous drinks. | [70] |
| astaxanthin | SLNs | Stearic acid; lecithin + poloxamer | Double emulsion solvent displacement | -; 6.11% (max) | The astaxanthin-SLN exhibited a sustained release behavior at pH 7.4; the intranasal administration of the SLN achieved higher biodistribution in the brain (than the intravenous route); it had an antioxidant potential against oxidative stress in neurological disorders. | [71] |
| Astaxanthin (oleoresin; 40 wt%) | NLCs | Glyceryl behenate; oleic acid; Tween 80 + lecithin | Hot shear homogenization + ultrasonication | ~90%; 1.8% | NLCs containing astaxanthin have a potential to be used in beverage formulations. | [72] |
| Astaxanthin | NLCs | Glyceryl palmitostearate; sunflower oil; Poloxamer | Hot homogenization | ~90%; ~1.2% | The NLCs formulation enhanced the antioxidant capacity of astaxanthin. | [73] |

| Liposoluble Nutraceuticals | Type of Lipid Nanoparticles [a] | Formulation Ingredients [b] | Processing Technique [c] | EE/DL [d] | Main Results | References |
|---|---|---|---|---|---|---|
| bixin | Polymer-coated SLNs | Capric/caprylic triglycerides; sorbitan monostearate; Tween 80 | Spontaneous emulsification | 100%; - | Lipid core nanocapsules with high EE (100%) of bixin and good physical stability were fabricated; the nanoencapsulation increased the stability of bixin against photosensitization and oxidation. | [74,75] |
| bixin | SLNs | Trimyristin or glycerol monostearate; lecithin + poloxamer 188 | Hot shear homogenization + ultrasonication | >99%; 6~18% | The release of bixin from the SLNs at pH 7.7 was of non-Fick diffusion; the oral administration of the bixin-SLNs resulted in enhanced in vivo hepatoprotection in rats. | [76] |
| Liposoluble vitamins | | | | | | |
| vitamin A (retinol) | SLNs | Glyceryl behenate; Tween 80 | Hot HPH | ~100%; 3.3% | The encapsulated retinol in SLNs displayed a controlled release behavior; the increased retinol release was correlated with polymorphic transitions of the lipids. | [77] |
| vitamin $D_2$ | SLNs | Tripalmitin; Tween 20 | Hot HPH | -; 15% (max) | Vitamin D-SLNs with a high LC (up to 15%) were successfully fabricated; increasing the vitamin proportion led to a progressive decrease in particle size of SLNs; the turbidity of the SLN dispersions reduced with increasing the loading. | [78] |

**Table 1.** *Cont.*

| Liposoluble Nutraceuticals | Type of Lipid Nanoparticles [a] | Formulation Ingredients [b] | Processing Technique [c] | EE/DL [d] | Main Results | References |
|---|---|---|---|---|---|---|
| vitamin $D_3$ | SLNs | Stearic acid + beewax; SDS | Solvent emulsification/evaporation | 43–78%; 19–350% | SLNs with extraordinary LC of vitamin D were successfully fabricated; increasing beewax ratio in the lipid matrix improved the encapsulation and release performance of as-obtained SLNs as nanovehicles for vitamin D; the SLNs were not cytotoxic and immunocompatible. | [79] |
| vitamin $D_3$ | NLCs | GMS; oleic acid; Tween 80 | Hot HPH | 68–86%; - | NLCs as nanovehicles for vitamin D were fabricated with widely available ingredients; the encapsulated vitamin D exhibited high stability upon storage or digestion and good controlled release behavior. | [80] |
| vitamin $D_3$ | NLCs | Glyceryldistearate or glyceryldibehenate; caprylic/caprictriglycerides or octyloctanoat; Tween 80, Tween 20 or poloxamer | Hot shear homogenization | -; - | The formulation of NLCs as nanovehicles for vitamin D was optimized; the incorporation of the vitamin D into NLCs increased its oral absorption. | [81] |
| vitamin $D_3$ | NLCs | GMS or polyglycerol polyricinoleate (PGPR); MCT or vegetable oils; poloxamer | Emulsification/evaporation | 85–92%; - | The NLCs with PGPR as the solid lipid exhibited higher liquid dispersion stability than those with GMS. | [82] |
| $\gamma$-tocotrienol | SLNs | Glyceryl behenate; poloxamer 188 | Hot shear homogenization + ultrasonication | -; 0.2% | The fabricated SLNs had much higher permeability and cell uptake than mixed micelles; the encapsulated $\gamma$-tocotrienol in the SLNs had threefold higher in vivo oral bioavailability. | [83] |

| Liposoluble Nutraceuticals | Type of Lipid Nanoparticles [a] | Formulation Ingredients [b] | Processing Technique [c] | EE/DL [d] | Main Results | References |
|---|---|---|---|---|---|---|
| $\alpha$-tocotrienol | SLNs | Glyceryl behenate; lecithin + poloxamer | Hot HPH | 58.5~82%; - | The formulation of the SLNs was optimized; the preparation at the optimal conditions was stable after 21 days of storage at 6 °C; the solid lipids in the SLNs were mainly present in the $\alpha$ and $\beta'$ polymorphic forms. | [84] |
| $\alpha$-tocotrienol | NLCs | Tristearin, MCT; poloxamer 188 | Hot shear homogenization + ultrasonication | 82.6%; 4.13% | The loading of the bioactive in the NLCs exhibited a lower toxicity on human cultured cells. | [85] |
| $\omega$-3 fatty acids | | | | | | |
| $\omega$-3 fatty acid-rich fish oil | SLNs, NLCs | Tripalmitin, fish oil; Tween 20 | Hot HPH | -; - | Increasing the fish oil in the lipid matrix (>10%) increased the stability of SLNs to aggregation, as well as the rate of $\alpha$- to $\beta$-polymorphic transitions of the solid lipid. | [86] |
| Docosahexaenoic acid (DHA) and $\alpha$-linolenic acid (ALA) | SLNs | Tripalmitin, tristearin, triolein; Tween 80 | Hot homogenization + microfluidization | -; - | The thermal behavior of the lipid matrix of SLNs could be modulated by changing the lipid components; the incorporation in the SLNs improved the oxidative stability and shelf life of DHA or ALA. | [87] |
| $\omega$-3 fatty acid-rich fish oil | SLNs, NLCs | Tristearin; low-(LM) or high-melting (HM) lecithins, taurodeoxycholate | Hot shear homogenization + microfluidization | -; - | The use of HM lecithin led to significant inhibition of oxidation of $\omega$-3 fatty acids in the NLCs, as a result of the formation of solidified surfactant shell layer. | [30] |
| $\omega$-3 fatty acid-rich krill oil | NLCs | Palm stearin; lecithin | Hot shear homogenization + ultrasonication | >96%; - | NLCs with small sizes and high loading efficiency were successfully fabricated using krill oil as the liquid lipid; the NLCs provided a good protection to the encapsulated bioactives against UV, and showed good physical and chemical stabilities upon long-term storage. | [88] |

**Table 1.** *Cont.*

| Liposoluble Nutraceuticals | Type of Lipid Nanoparticles [a] | Formulation Ingredients [b] | Processing Technique [c] | EE/DL [d] | Main Results | References |
|---|---|---|---|---|---|---|
| Linseed oil + quercetin | NLCs | GMS; linseed oil (liquid lipid); Tween 80 | Hot HPH | -; - | Both quercetin and linseed oil were successfully co-loaded in NLCs with better lipid oxidation or storage stability; the addition of linseed oil increased the antioxidant capacities of quercetin. | [89] |
| Conjugated linoleic acid | NLCs | Cocoa butter; conjugated linoleic acid; Poloxamer | Hot high-shear homogenization | -; 98.2% | The formation of NLCs greatly protected conjugated linoleic acid against oxidation and heating. | [90] |
| Essential oils | | | | | | |
| citral | NLCs | Glyceryl palmitostearate; MCT; poloxamer | Hot shear homogenization | 99.8%; 12.5% | The encapsulation in NLCs provided prolonged preservative effect and storage stability. | [91] |
| citral | SLNs | GMS; Tween 80 + Span 80 | Hot HPH | 48–73%; 22.8% | The incorporation of citral in the SLNs reduced the ordered crystallinity of GMS; the nanoencapsulation significantly improved the retention of citral upon 12 days of storage. | [92] |
| eugenol | NLCs | Tristearin, MCT; poloxamer 188 | Hot shear homogenization + ultrasonication | 81.4%; 3.92% | Better antimicrobial activity and a lower toxicity on human cultured cells were demonstrated for encapsulated eugenol in the NLCs | [85] |
| carvacrol | SLNs | Propylene glycol monopalmitate, glyceryl monostearate; Tween 80 | Microemulsion template method | >98%; up to 30% | Carvacrol was homogenously distributed within the SLNs; the encapsulated carvacrol exhibited more effective anti-microbial activites. | [93] |
| carvacrol | SLNs | Beeswax; Tween 80 + lecithin | Hot shear homogenization + ultrasonication | 88.5%; - | Both carvacrol and astaxanthin were successfully co-loaded in the SLNs with high EE; encapsulated bioactives were stable under oxidative, acidic and alkaline conditions, and showed better anti-microbial activities. | [94] |

**Table 1.** *Cont.*

| Liposoluble Nutraceuticals | Type of Lipid Nanoparticles [a] | Formulation Ingredients [b] | Processing Technique [c] | EE/DL [d] | Main Results | References |
|---|---|---|---|---|---|---|
| Frankincense and myrrh essential oil | SLNs | Glyceryl dibehenate/behenate; lecithin + Tween 80 | Hot HPH | 80.6%; ~53.7% | The encapsulation of the oil decreased the ordered crystallization of solid lipids in the SLNs, and significantly improved its antitumor efficacy in mice. | [95] |
| Other liposoluble bioactives | | | | | | |
| $CoQ_{10}$ | SLNs | Cetyl palmitate; Tego Care 450 | Hot HPH | -; 2.4% | The majority of $CoQ_{10}$ was homogenously mixed with the solid lipid matrix, while the others formed separate domains. | [47] |

[a] SLNs: solid lipid nanoparticles; NLCs: nanostructured lipid carriers; [b] GDS: glycerol distearate; GMS: glycerol monostearate; MCT: medium chain triglycerides (e.g., Miglyol ®810); [c] HPH: high pressure homogenization. [d] EE: encapsulation efficiency (%); DL: drug loading (wt%; relative to total lipid content).

*4.1. Liposoluble Nutraceuticals*

4.1.1. Carotenoids

Carotenoids are important isoprenoid pigments (from a yellow to red spectrum distribution), usually synthesized by either photosynthetic organisms or non-photosynthetic organisms (e.g., some fungi or bacteria) [96]. They are valuable liposoluble bioactive compounds with a lot of well-recognized beneficial effects on human health, including pro-vitamin A function (some of them), improvement in eye and heart health, enhancement of cognitive functions, prevention of skin cancer (or providing UV protection), immunomodulation/stimulation and reductions in risk of cardiovascular disease, cataracts, and age-related macular degeneration [9]. Most health effects of carotenoids have been associated with their potent antioxidant properties, including their free radical-quenching capacity [97]. $\beta$-Carotene can be converted into vitamin A in the body, while zeaxanthin and lutein seem to mainly act as antioxidants, e.g., scavenging free radicals generated in the biological system, and absorbing sunlight with particular wavelengths in the eye [97]. The chemical structure and properties of carotenoids considerably vary. Carotenoids are generally categorized into two main subgroups according to their structure and properties [9,96,97]. See below:

*Nonpolar carotenes*: These carotenoids have a hydrocarbon ring at one or both ends of the polyene backbone, having no oxygen or polar functional groups. They are highly liposoluble and can be accumulated in LDL or localized within the hydrophobic center of the plasma membrane. They can be further divided into two groups: pro-vitamin A (e.g., $\beta$-carotene and $\beta$-cryptoxanthin) and non-pro-vitamin A (e.g., lutein or lycopene).

*Polar xanthophylls/phylloxanthins*: These are carotene derivatives with oxygen-containing groups, e.g., -OH (lutein), =O (canthaxanthin), a combination of -OH and =O (astaxanthin), or asters of alcohol (fucoxanthin). Xanthophylls are stronger in polarity than their parental counterparts.

Carotenoids, as ingredients or additives in the food industry, are generally susceptible to light (especially UV light), oxygen, and heat [9]. The lipophilic nature of carotenoids possessing several conjugated double bonds in their polyene backbone makes them very poorly soluble in water, thus leading to low bioavailability. Thus, their ease of oxidation degradation, very low water solubility, and low bioavailability are the main factors limiting their application in food or pharmaceutical formulations. Both SLNs and NLCs have been successfully developed and utilized as safe and attractive nanovehicles for overcoming the above limitations of carotenoids [9,98,99]. Table 1 summarizes many previous works addressing the development and utilization of SLNs and/or NLCs as nanovehicles for different carotenoids, including $\beta$-carotene, lutein, lycopene, astaxanthin, and bixin. There are basically two kinds of commercial carotenoid products used in the formulation of SLNs and NLCs: high-purity carotenoid crystals and carotenoid oleoresins. Considering that the main components in many carotenoid oleoresins are liquid vegetable oils, this kind of carotenoid product can be directly used as the liquid lipid to fabricate NLCs as nanovehicles for carotenoids.

$\beta$-Carotene is one of the most important bioactive carotenoids. It has been relatively widely investigated for its potential to be incorporated in SLNs and/or NLCs to facilitate better stability (against chemical degradation) [46,49–51,63,64]. In these cases, most SLNs and NLCs can be fabricated, with Tweens (20–80) as the emulsifiers, by a melt-emulsification with high pressure homogenization (HPH) or ultrasonication, followed by a cooling-induced solidification process (Table 1). In an early work, Hentschel et al. [49] directly applied the $\beta$-carotene dispersion (10% in sunflower oil) as the liquid lipid to fabricate progylene glycol monostearate (PGMS) NLCs as nanovehicles for $\beta$-carotene, with the aim of evaluating the feasibility of utilizing $\beta$-carotene-NLCs as functional ingredients in beverages. The total lipid content of the formed NLC dispersions was 10%, with the liquid lipid ratio (relative to total lipid content) increasing up to 4.5%. The results indicated that the $\beta$-carotene-NLC dispersions were stable after storage for 30 weeks at 4–8 °C, but the encapsulated $\beta$-carotene suffered from distinct oxidation degradation upon prolonged

storage (e.g., >10 days). The $\beta$-carotene stability could be greatly improved by the co-incorporation of $\alpha$-tocopherol in the lipid phase. A similar enhancement of chemical stability of $\beta$-carotene was also reported for NLCs fabricated with other lipid components or using different fabrication processes [46,63,64], and confirmed in SLNs [50,51].

The encapsulation performance of $\beta$-carotene in SLNs and NLCs has not been thoroughly evaluated in the literature. In several cases of purified $\beta$-carotene to be encapsulated in NLCs, the EE of $\beta$-carotene considerably varied (from 16–100%) depending on the relative ratio of the liquid lipid and temperature [46,63]. In NLCs fabricated using the solvent diffusion method, it was indicated that the oxidative degradation of encapsulated $\beta$-carotene was affected more significantly by the liquid lipid to total lipid ratio and temperature than the total lipid concentration and surfactant concentration [63]. Pan et al. [46], interestingly, indicated that, in NLCs stabilized by high-melting lecithin, the $\beta$-carotene loading efficiency progressively increased from ~16% to 70%, as the liquid lipid to total lipid ratio increased from 0% to 70%. Herein, an issue of $\beta$-carotene exclusion from the solid lipid matrix should be taken into account. Pan et al. [46] demonstrated that, as the liquid lipid to total lipid ratio increased, the extent of $\beta$-carotene exclusion decreased, while the distribution uniformity in the lipid matrix increased. Additionally, the oxidative stability of encapsulated $\beta$-carotene was negatively related to the extent of the $\beta$-carotene exclusion. Impaired oxidative stability in encapsulated $\beta$-carotene by $\beta$-carotene exclusion from the solid lipid matrix has also been observed in SLNs, as compared with liquid lipid nanoparticles [51]. Furthermore, Helgason et al. [50] found that the use of high-melting lecithin as the surfactant could provide better protection against chemical degradation in encapsulated $\beta$-carotene in SLNs than low-melting lecithin. The observed better protection against $\beta$-carotene degradation, in the high-melting lecithin case, was attributed to the retarded polymorphic transition ($\alpha \rightarrow \beta'$ or $\beta$) of solid lipid crystals [50]. Perhaps, due to the occurrence of $\beta$-carotene exclusion in both SLNs and NLCs, the loading amount (LA) of $\beta$-carotene was generally low (0.1%–0.5%; relative to total lipid content) (Table 1). In a recent work addressing the in vitro digestibility and $\beta$-carotene bioaccessibility of SLNs, it was, interestingly, indicated that there was no direct relationship between the in vitro digestibility of solid lipids and $\beta$-carotene bioaccessibility [65].

Lutein is another important carotenoid, typically derived from different vegetables (e.g., spinach) and flowers. The commercial lutein product mainly comes from a few limited sources, including marigold flowers. The solubility of this carotenoid in aqueous media is very low, thus exhibiting poor oral-bioavailability; moreover, it is very unstable and apt to chemical degradation [9]. Few works have successfully developed NLCs as nanovehicles for improving the stability and sustained release behavior of lutein [59,66,67]. Astonishingly, all such works applied the same commercial lutein product (20%, in corn oil) from the same company, confirming the difficulty of utilizing this carotenoid in food and pharmaceutical formulations. A maximum LC of about 3% (relative to total lipid content) with EE of ~85% was obtained for NLCs fabricated by hot ultrasonication using lutein oleoresin (20%, in corn oil) as the liquid lipid [66]. The size of fabricated lutein-NLCs was dependent on the ultrasonication duration and the lutein loading amount. It was also confirmed that the incorporation of the lutein oleoresin led to the formation of an imperfect crystalline lattice [66], which was favorable for lutein loading in the lipid matrix. Lutein NLCs can also be fabricated using medium-chain triglycerides (MCT) or $\omega$-3 fatty acids as the liquid lipid, and in these cases, the lutein oleoresin was well mixed with the liquid lipid [59,67]. The encapsulated lutein in these nanoparticles exhibited better protection against UV degradation and greater free radical scavenging ability, as well as a more sustained release behavior (Table 1).

In more recent years, interest in the development of SLNs or NLCs as nanovehicles for lycopene, astaxanthin and bixin has increased [68–76,99]. These carotenoids possess many outstanding beneficial health effects, exhibiting great potential for application in the formulations of health-related products. Lycopene, widely present in tomato, watermelon, and pink grapefruit, is one of the most potent natural antioxidants [99]. It is an acyclic

open-chain unsaturated carotenoid with 11 conjugated double bonds, having a very high lipophilicity (with a log *p* value of 17.64). This carotenoid is insoluble in water (its solubility in oil at ambient temperatures is also low) and sensitive to oxidants, light, and heat, thus leading to low bioavailability [70]. Okonogi and Riangjanapatee [68] selected orange wax and rice bran oil (with great potential to solubilize lycopene) as the solid and liquid lipids, and successfully fabricated NLCs as nanovehicles for lycopene via a hot HPH method, with a very high EE of 100% and LC values of 0.1–1.0% (relative to total lipid content). The encapsulation in the NLCs improved the chemical stability against degradation of lycopene after long-term storage, up to 3 months [68]. Using a similar fabrication process, Nazemiyeh et al. [69] and Zardini et al. [70] implemented the high-performance nanoencapsulation of lycopene in SLNs and/or NLCs, with maximum EE and LC values reaching up to 98% and 5.5%, respectively. As expected, the EE of NLCs was significantly higher than that of SLNs [70]. The feasibility of such lycopene-NLCs or -SLNs to be applied in orange drinks was explored, and it was indicated that the addition of encapsulated lycopene facilitated the solubilization of this nutraceutical in aqueous samples [70]. Bixin is another naturally occurring carotenoid. It has extremely low aqueous solubility, but good solubility in oil [9]. Utilizing the high potential of bixin to be well mixed with the lipid phase, Rao et al. [76] efficiently encapsulated this carotenoid in SLNs using a hot melt-emulsification process, with EE > 99% and maximum LC value up to 18% (relative to total lipid content). They demonstrated that the nanoencapsulation of bixin in the SLNs significantly increased the in vivo hepatoprotection against paracetamol-induced injury in animal experiments. The high EE encapsulation of bixin was confirmed in polymer-coated SLNs fabricated through a spontaneous emulsification process [74,75], wherein it was also confirmed that the nanoencapsulation improved the photo- and oxidative stability of bixin.

Astaxanthin is a xanthophyll carotenoid, found in aquatic animals (e.g., crabs and salmon) and mainly produced by microalgae [71]. This carotenoid shows a free radical scavenging activity 100 times more potent than vitamin E, and 10 times more than $\beta$-carotene. It also exhibits great potential for use in preventing and treating oxidative stress-related diseases [71], thus having a high commercial value. Like other carotenoids, astaxanthin has low solubility in water, easy degradation, and low bioavailability. Astaxanthin oleoresin (40 wt%), or astaxanthin extract solubilized in sunflower oil, has been successfully incorporated in NLCs with oleic acid or sunflower oil as the liquid lipid using the hot melt-emulsification process [72,73]. The encapsulation of astaxanthin (in oleoresin form or in sunflower oil) within the NLCs had similar EE and LC values of ~90% and 1.2–1.8% (relative to total lipid content), respectively (Table 1). The nanoencapsulation in NLCs enhanced the antioxidant capacity of astaxanthin [73]. Bhatt et al. [71] successfully fabricated SLNs using a double emulsion solvent displacement method, with a much higher LC of 6.11%. They demonstrated that the encapsulated astaxanthin could provide protection against oxidative stress in neurological disorders.

### 4.1.2. Fat-Soluble Vitamins

Vitamins are important micronutrients. They are essential to human nutrition and health, and play roles in normal metabolism, cellular regulation, growth, and development [100,101]. There are two kinds of vitamins: water-soluble and fat-soluble. The fat-soluble vitamins include vitamin A (retinol, retinal and retinyl esters), vitamin $D_2$ (ergocalciferol), vitamin $D_3$ (cholecalciferol), vitamin E (tocopherol) and vitamins $K_1$ (phylloquinone) or $K_2$ (menaquinone). These liposoluble vitamins cannot be readily solubilized in the gastrointestinal tract, and are apt to chemical degradation under oxidative, pH, light, and heat stresses, thus exhibiting low bioavailability [100]. The use of lipid nanoparticles (SLNs and NLCs in particular) as oral nanovehicles for such liposoluble vitamins has been considered an effective strategy for improving their solubility in aqueous phase, chemical stability, epithelium permeability, and bioavailability [100]. In this regard, the EE for liposoluble vitamins (e.g., vitamin A or D) to be incorporated in such lipid nanovehicles can

be close to 100% [77,102,103]. This is consistent with the fact that all liposoluble vitamins are very biocompatible and miscible with the lipid matrix in SLNs and/or NLCs.

Vitamin D is an important fat-soluble micronutrient or nutraceutical. It mainly exists in two forms: i) vitamin $D_2$ (ergocalciferol), synthesized only by plants, and ii) vitamin $D_3$ (cholecalciferol), synthesized by the human body. In addition to the traditional role in the human bone metabolism, vitamin D is of crucial importance for the prevention of many diseases, e.g., neurodegenerative diseases, cardiovascular diseases, diabetes, and cancers [104]. The insufficient intake and deficiency of this vitamin is still a global issue, especially for infants, children, and child-bearing women [104]. As expected, both SLNs and NLCs can be applied to act as nanovehicles for vitamin D ($D_2$ or $D_3$) [78–82]. The EE and LC of vitamin D in these nanoparticles are highly variable with the applied lipid composition (and type of lipid nanoparticles) and emulsification process (Table 1). Patel and Martin-Gonzalez [78] evaluated the effectiveness of using tripalmitin SLNs stabilized by Tween 20 as nanovehicles for vitamin $D_2$. The SLN dispersions (5%, $w/w$) with the vitamin incorporated in the molten lipid were formed by an HPH process at 80 °C. In this work, the EE was undetermined, while the maximal LC was able to reach 15% (relative to the lipid content). Interestingly, it was indicated that increasing the vitamin-to-lipid ratio from 0% to 20% resulted in a progressive decrease, from 120 to about 65 nm, in $z$-average diameter of the SLNs. Accordingly, the turbidity of the dispersions were noticeably reduced [78]. The findings confirmed the high capacity of SLNs to deliver this vitamin, thus providing an alternative to milk or margarine as a source of vitamin D. If SLNs are formed using a combined solvent emulsification and evaporation strategy, instead of HPH, the LC of vitamin D in the nanoparticles can be even higher. Demirbilek et al. [79] reported that SLNs of stearic acid and beeswax blend, fabricated using the solvent emulsification and evaporation approach, exhibited an LC of 19%–350% for vitamin $D_3$ incorporation, though the EE was only in the range 43–78%. They indicated that increasing the beeswax ratio in the lipid matrix decreased the storage stability of encapsulated vitamin $D_3$, but increased its EE and release rate (in the phosphate buffer). Park et al. [80] successfully fabricated NLCs consisting of a glycerol monostearate (solid lipid) and oleic acid (liquid lipid) blend as nanovehicles for improving the stability and release behavior of vitamin $D_3$, using an HPH process. The as-fabricated NLCs had high EE values of 68–86%, with the EE decreasing with increasing vitamin D concentration. The incorporation of the vitamin D in the NLCs remarkably improved its stability over long-term storage and during simulated gastric digestion, as well as its release behavior during digestion [80]. The stability of vitamin $D_3$-loaded NLC dispersion was more related to the choice of emulsifier than the use of different carrier oils [82]. The vitamin D-loaded NLCs, or lipid nanocapsules, were very compatible with food formulations, e.g., milk beverages [102,105], thus representing great potential in food applications.

As expected, the incorporation of vitamin $D_3$ in the NLCs enhanced its in vivo absorption upon oral administration [81]. In addition to the enhanced bioavailability, the incorporation of vitamin D in NLCs can provide a colon-targeted delivery via oral administration. Zai and colleagues [106] confirmed that, when orally administrated to mice, vitamin $D_3$ in the NLCs was delivered to the colon, and remained in the colonic tract at a high concentration for a long period. More interestingly, the oral administration of vitamin $D_3$-NLCs showed multiple effects on the suppression of symptoms of colitis induced by dextran sodium sulfate (DSS) [106]. In a more recent work, it was reported that the oral administration of a vitamin $D_3$-NLC ameliorated the non-alcoholic steatohepatitis progression in mice via enhanced intestinal barrier support and suppressed inflammation and fibrosis in the liver [107]. In concurrent administration with doxorubicin (an anticancer agent), the application of vitamin $D_3$-loaded NLC was shown to greatly elevate the efficacy of the anticancer activities of doxorubicin in breast cancer [108].

Vitamin E is another important liposoluble vitamin that occurs naturally in vegetable oils and deodorizer distillates. This vitamin is well recognized for its role as an antioxidant, scavenging lipid peroxyl radicals or acting as a synergist with other antioxidants [109].

Vitamin E is, in fact, a group, consisting of eight different tocol compounds, including tocopherols and tocotrienols. In addition to the direct oxidative protection, vitamin E exhibits a variety of oxidative stress-related health-benefitting effects on cardiovascular diseases, cancer, immune response, inflammatory diseases, neurological disorders, cataracts, and age-related macular degeneration [110,111]. Despite these beneficial effects, all tocopherols and tocotrienols are difficult to incorporate in food formulations, due to their insolubility in water. A few works have reported that the incorporation of vitamin E in SLNs or NLCs not only facilitated its application in food formulations, but also increased its oral bioavailability and storage stability [83–85]. In these works, both the SLNs and NLCs were fabricated using the melt-emulsification process (e.g., hot HPH), using poloxamer as the emulsifier, alone or in combination with lecithin. The EE and LC also considerably varied with the type and composition of lipid nanoparticles and homogenization conditions (Table 1).

4.1.3. Omega-3 Polyunsaturated Fatty Acids ($\omega$-3 PUFAs)

The very long chain $\omega$-3 PUFAs, e.g., eicosapentaenoic acid (EPA) and docosahexaenoic acid (DHA), in the form of fish oils and $\alpha$-linolenic acid (ALA) from flaxseed oil, have been widely accepted as important parts of modern human nutrition, thanks to their health-promoting effects on cardiovascular diseases, inflammatory diseases, diseases in metabolism, diabetes, cancers, brain diseases, and neurological disorders [112,113]. The $\omega$-3 PUFAs are not synthesized in the body, and should be included in the diet for human nutrition and health. The supplementation with $\omega$-3 PUFAs alters the fatty acid composition of plasma, cells, and tissues in humans, and an increased intake of $\omega$-3 PUFAs is known to exert beneficial effects on human health [114]. However, the application of $\omega$-3 PUFAs in food formulations is limited by their unpleasant flavors, a result of high oxidative instability [19]. Many effective strategies have been proposed and developed to protect $\omega$-3 PUFAs from oxidation during production, storage, transport, and consumption. Because $\omega$-3 PUFAs or $\omega$-3 PUFA-rich oils are liquid lipids, lipid nanoparticles (and NLCs in particular) can be considered among the most appropriate and promising nanovehicles for these bioactive ingredients.

$\omega$-3 PUFAs can readily mix with melted solid lipids at any relative ratio, as required. Hot HPH, or hot homogenization, in combination with ultrasonication, is generally used as the emulsification process to fabricate SLNs or NLCs for $\omega$-3 PUFA incorporation (Table 1). Tweens or lecithins are good emulsifiers for the formation and stabilization of such nanovehicles. If the relative ratio of added $\omega$-3 PUFAs is low, they are generally evenly distributed in the solid lipid matrix upon cooling/solidification. Awad and colleagues [86] indicated that the crystallization, polymorphic transformation, and stability of tripalmitin SLNs were highly affected by the addition of $\omega$-3 PUFA-rich fish oil at different ratios (relative to solid lipid content). In the absence of fish oil, the SLN suspensions tended to form a gel after the tripalmitin crystallization, attributed to the aggregation and network formation as a result of particle shape changes. The addition of fish oil to the melted lipid at a ratio of 10 wt% or above increased the stability of the corresponding SLNs to aggregation [86]. Further evidence showed that, upon increasing the fish oil ratio, the rate of $\alpha$- to $\beta$-polymorphic transitions of tripalmitin increased, and the formed crystals grew less ordered [86]. Holser [87] confirmed that the encapsulation of DHA and ALA (in the form of esters) in SLNs consisting of triglycerides protected such $\omega$-3 PUFAs from oxidation and improved the shelf life of the corresponding formulated products. In fact, the incorporated $\omega$-3 PUFAs could be directly applied as the liquid lipid for the fabrication of NLCs. For example, NLCs using palm stearin and krill oil as the solid and liquid lipids had a less-ordered crystalline structure, thus exhibiting a high EE (>96%) and LC of $\omega$-3 PUFAs [88]. Such NLCs were demonstrated to offer good protection against UV exposure to encapsulated $\omega$-3 PUFAs, as well as good physical and chemical stability for long-term storage [88]. They also offered good protection to encapsulated conjugated linoleic acid against oxidation and heating [90]. Interestingly, NLCs fabricated using ALA-rich linseed oil as the liquid lipid could also act as nanovehicles for the encapsulation and delivery of

poorly soluble bioactives (e.g., quercetin) and, in this case, the nano-encapsulated quercetin tended to be present in the NLCs with less ordered lipid cores (or with more relative ratios of linseed oil) [89]. The co-loading of quercetin and linseed oil not only improved the in vitro antioxidant capacities of quercetin-loaded NLCs, but also increased the oxidation stabilities of lipids. It also increased physical stability upon long-term storage [89].

Salminen et al. [30] indicated that the oxidative and physical stability of fish oil-loaded NLCs stabilized by lecithin was highly dependent on the type of lecithin used (high- or low-melting). They observed that the NLCs stabilized with a high-melting lecithin exhibited much greater stability against oxidation of $\omega$-3 PUFAs in fish oil than those with low-melting lecithin. This improved stability was attributed to the formation of solidified shell layer, induced by the crystallized high-melting lecithin as the interfacial heterogeneous nuclei [30]. To better illustrate this, they proposed a mechanistic explanation for the formation of core-shell lipid nanoparticles stabilized by the high-melting surfactant (lecithin), as displayed in Figure 6. Upon cooling to 35 °C, the adsorbed high-melting lecithin (80H) on the interface of emulsified droplets (consisting of a melted mixture of tristearin and fish oil) crystallized (solidified) before the high-melting lipid (tristearin). The crystallized 80H was subsequently able to act as the heterogenous nuclei and induce crystallization of tristearin close to the 80H solid surface. When the temperature was further cooled to 20 °C, interfacial heterogenous nucleation led to the formation of an ordered crystal structure, with lipid liquid (fish oil) incorporated within the solid lipid matrix (Figure 6). Meanwhile, in the case of low-melting lecithin (PC 75), PC 75, at the interface of emulsion droplets, remained liquid upon cooling to 35 °C, and as a result, no crystallization of tristearin was induced. When the temperature was decreased to 20 °C, the melted tristearin crystallized to form a solid lipid matrix, leading to the occurrence of expulsion of $\omega$-3 PUFAs from the crystallized lipid matrix (Figure 6). The excluded $\omega$-3 PUFAs from the solid matrix seemed to be prone to chemical oxidation. It should be noted that, aside from the use of a high-melting surfactant, core-shell NLCs can also be formed by additional coating with a charged polymer on the NLCs. Rabelo et al. [103] successfully prepared chitosan-coated NLCs, consisting of stearic and oleic acids as the solid and liquid lipids, respectively, by electrostatic deposition, and confirmed that the coating greatly increased the physical stability of the as-fabricated NLCs. However, at the time of this writing, whether the oxidative stability of liquid lipids in NLCs can be improved remains to be seen.

### 4.1.4. Essential Oils

Essential oils are volatile flavor substances extracted from plant tissues. They exhibit a high potential to inhibit the growth of microorganisms [115], but they are chemically unstable, prone to oxidation and volatilization, which limits their application in food preservation [116]. Encapsulation is a popular technique in the field to stabilize and release flavor compounds sustainably [117]. Among all the reported encapsulation techniques, lipid-based nanovehicles, such as nanoemulsions, SLNs, and NLCs, seem to be among the most promising (and most investigated) delivery systems for antimicrobials, including essential oils [21]. To date, however, only nanoemulsions, liposomes, and biopolymeric particles have been successfully applied in real foods [116]. Table 1 also displays some examples of using SLNs or NLCs as containers or sustained-release delivery systems for essential oils (including citral, eugenol, and carvacrol). The EE of essential oils in such lipid nanoparticles ranged from 48 to > 98%, depending on the type and addition ratio (relative to solid lipid content) of essential oils, and fabrication process (Table 1), reflecting the volatile and unstable nature of these hydrophobic flavor substances during the processing. The maximal LC could reach up to 30% for carvacrol, to be loaded in SLNs formed using the microemulsion template method [93], or up to approximately 53.7% for frankincense and myrrh essential oils, to be loaded in SLNs using the hot HPH process [95].

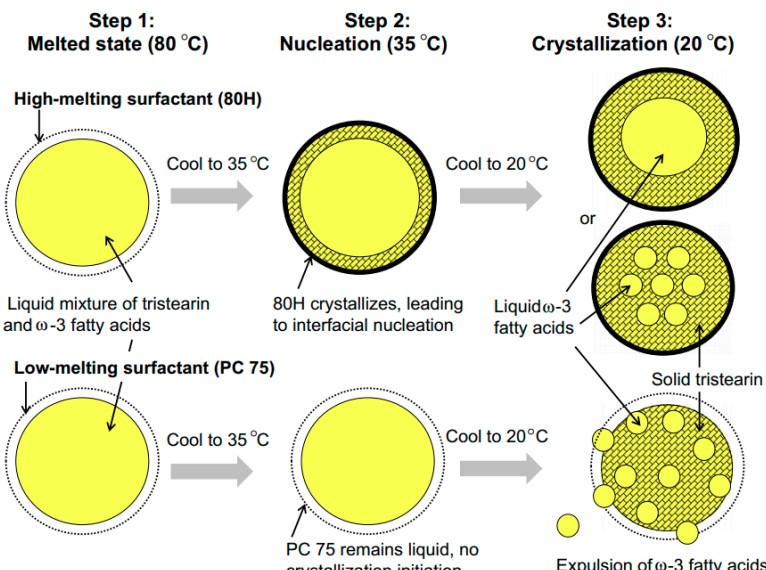

**Figure 6.** Schematic illustration for the influence of high-melting and low-melting surfactants on the nucleation and crystallization of lipid nanoparticles. Using high-melting lecithin (80H) as the surfactant, core-shell lipid nanoparticles, with $\omega$-3 PUFAs as the cores, are formed, while, in the case of low-melting lecithin (PC75), a fraction of $\omega$-3 PUFAs might exclude from the crystallized solid lipid matrix upon cooling. (Adapted from [30]).

As in the case of fish oil-loaded NLCs, many essential oils can be directly used as the liquid lipid to fabricate NLCs as nanovehicles for themselves. For example, Mokarizadeh et al. [91] successfully fabricated such NLCs as nanovehicles for citral, with a very high EE of 99.8%, and showed that the loaded citral in the NLCs was more effective as a natural preservative agent for the inhibition of various gram-positive or negative bacteria and fungi, than the conventional citral emulsion. Citral could be also evenly distributed in the solid lipid matrix, even at a high LC of 22.8% and, as expected, the incorporation of citral reduced the crystallinity of the solid lipid (glyceryl monostearate) and improved the storage stability of citral [92]. A similar result of a homogenous distribution was observed for the carvacrol-loaded SLNs formed though a microemulsion template method, at a very high LC of up to 30% [93].

Shakeri and colleagues [109] reported that astaxanthin could be co-entrapped with carvacrol in the SLNs with beeswax as the solid lipid, at a high EE of 88.5% (for carvacrol). Most of these previous works indicated that the nanoencapsulation of such essential oils (or natural antimicrobials) in SLNs or NLCs improved their antimicrobial activities, but did not show distinct toxicity on human-cultured cells [26,85,91,93,109,116].

### 4.2. Poorly Soluble Bioactive Compounds

#### 4.2.1. Curcuminoids (and Curcumin in Particular)

Curcuminoids are the bioactive components of rhizomes of turmeric (*Curcuma longa* L.; Figure 7a), accounting for 2~6% of dried rhizomes. Curcuminoids from turmeric are typically a mixture of curcumin (52–63%), demethoxycurcumin (19–27%) and bisdemethoxy-curcumin (18–28%) (Figure 7b). Commercially-available dry turmeric extracts (standardized to 95% of curcuminoids) usually include curcumin (70–80%), demethoxycurcumin (15–25%) and bisdemethoxy-curcumin (2.5–6.5%) [118]. To date, curcuminoids in general (and curcumin in particular) have attracted extensive attention for their health benefits, including antioxidant and anti-inflammatory activities, neuroprotection, chemo- and cancer prevention, anti-cardiovascular disease, anti-obesity and anti-diabetic activities, and anti-infective properties [60,61,119]. Many animal and clinical studies have shown that curcumin is very safe, even at high doses (e.g., 12 g/day) in humans [120]. Despite all this, the bioavailability of curcumin is poor, due to its instability at physiological pH, insolubility in water, poor

absorption, rapid metabolism, and rapid systemic elimination [118,120], which greatly hinders its formulations in foods and pharmaceuticals.

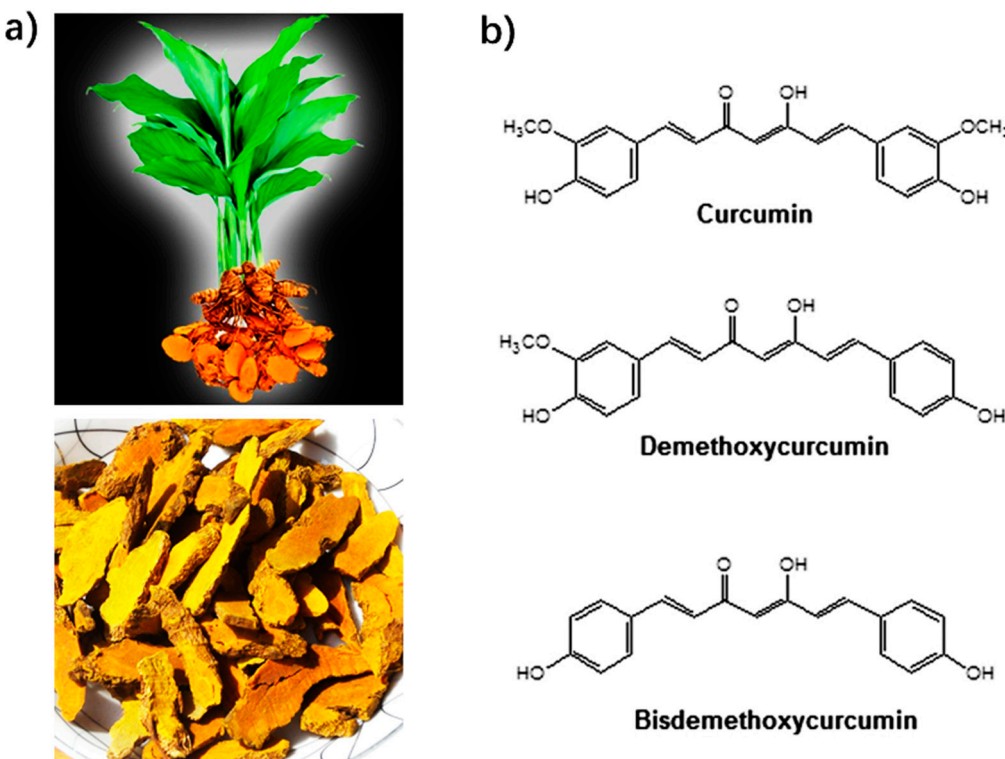

**Figure 7.** (**a**) Photographs of *curcuma longa* L. and its dried rhizomes. (**b**) Chemical structure of curcuminoids (curcumin, demethoxycurcumin and disdemethoxycurcumin) from turmeric.

During the past decade, great effort has gone toward facilitating use a variety of lipid nanoparticles (SLNs and NLCs) as nanovehicles for improving the stability, bioaccessibility and/or bioavailability, and bioactivities of curcumin. Considering that curcumin is a poorly soluble and unstable polyphenol in water, it is of crucial importance to decide when and how curcumin is introduced into the system for its optimized encapsulation in the SLNs or NLCs. There are two essential modes through which curcumin is incorporated into lipid nanoparticles. Curcumin or turmeric powder (or crystals) can be directly dissolved in the melted lipid phase at high temperatures of SLNs [89,121–124], or NLCs [52,55,125,126], or solubilized in an appropriate organic solvent prior to mixing with the melted lipid phase [32,35,54,127–129]. When an organic solvent is involved in the incorporation of curcumin, an additional evaporation is usually needed to remove the organic solvent, before or after the melt emulsification. Table 2 presents some representative examples of using SLNs and/or NLCs as delivery systems for curcumin, or curcuminoids (turmeric extract). In most cases, the SLNs and NLCs were fabricated using a melt emulsification process followed by a cooling-induced solidification, e.g., hot HPH (high pressure homogenization), or hot shear homogenization and ultrasonication (Table 2).

**Table 2.** A summary of some representative works addressing the fabrication and characterization of SLNs and/or NLCs as food-grade nanovehicles for curcuminoids or turmeric extract.

| Curcuminoids | Type of Lipid Nanoparticles [a] | Formulation Ingredients (Lipids and Emulsifiers) [b] | Processing Technique [c] | Mode of Drug Incorporation [d] | EE/DL [e] | Main Results | References |
|---|---|---|---|---|---|---|---|
| curcumin | SLNs | Compritol®888 ATO; Tween 80 + soy lecithin | Microemulsion template method | Dissolution (in hot emulsifier mix) | 82%; 10% | The in vitro release of encapsulated curcumin followed the diffusion pattern; the encapsulated curcumin in the SLNs exhibited good long-term stability, and high bioavailability (32–155 times as free curcumin) in rats. | [130] |
| curcuminoids | NLCs | MCT, refined castor oil, soybean oil, TM or TS; polomaxer | Hot HPH | Dissolution | ~97%; max. 0.1% | The encapsulated curcuminoids exhibited good long-term storage stability; their release from the nanoparticles was dependent on the applied medium and physical state of the lipid carrier, which was much more affected by the degradation of the lipid matrix. | [52] |
| curcumin | SLNS | Trimyristin; different emulsifiers | Hot HPH | dissolution | ~66%; ~0.05 wt% | The encapsulation of curcumin in SLNs showed enhanced delivery (compared to unencapsulated or emulsified curcumin); the transport route of the SLNs was simple diffusion; most of encapsulated curcumin was absorbed and metabolized by the cells. | [122] |
| Curcumin (+ genistein) | NLCs | Oleic acid (liquid), GMS (solid); Tween 80 | Hot shear homogenization + ultrasonication | Dissolution | >75%; 1.2% (alone) or 0.7% (in combination) | The co-loading with genistein increased the loading efficiency, and the inhibition against prostate cancer cells. | [131] |
| curcumin | SLNs | Compritol®888 ATO; sodium caseinate (NaCas) | Solvent-diffusion+ hot shear homogenization + ultrasonication | Mixing (in ethanol) | 40–80%; - | Novel SLNs with biopolymeric double layer coating (using NaCas and pectin) were successfully fabricated as nanovehicles for curcumin; the cross-linking of the layer coating improved the EE, DL, stability and release behavior, as well as the antioxidant activity of encapsulated curcumin in aqueous phase; the cross-linking further facilitated the spray drying of SLNs to form homogenous powders. | [129] |

**Table 2.** *Cont.*

| Curcuminoids | Type of Lipid Nanoparticles [a] | Formulation Ingredients (Lipids and Emulsifiers) [b] | Processing Technique [c] | Mode of Drug Incorporation [d] | EE/DL [e] | Main Results | References |
|---|---|---|---|---|---|---|---|
| curcumin | SLNs | Glycerol stearate, propylene glycol esters of fatty acids, palmitic acid; Tween 80 | Hot shear homogenization + ultrasonication | dissolution | 100%; 10% | The oral bioavailability of encapsulated curcumin in the SLNs was significantly improved by the coating with chitosan. | [124] |
| curcumin | SLNs, NLCs | Compritol®888 ATO (solid), oleic acid (liquid); sodium caseinate (NaCas) + Tween 80 (emulsifiers) | Solvent-diffusion + hot ultrasonication | Mixing (in an acetone and ethanol mixture) | 33–66%; max. 4.95% | SLNs and NLCs with high loading capacity and exceptional gastrointestinal stability were fabricated using NaCas (together with a minimal concentration of Tween 80) as the emulsifier and pectin as the coating, especially when the layer coating was crosslinked. | [132] |
| curcumin | NLCs | MCT (liquid), Compritol®888 ATO (solid); Tween 80 | Hot HPH | Mixing (in acetone) | -; - | The release of encapsulated curcumin from the NLCs was consistent with the release of free fatty acids, which could be modulated by altering the lipid type and composition, and the use of lipase inhibitors. | [54] |
| curcumin | SLNs | Stearic acid; NaCas (emulsifier) | Solvent-diffusion + hot ultrasonication | Mixing (in ethanol) or dissolution | -; - | The influence of loading processes on the efficacy of encapsulation of curcumin in the SLNs was investigated; the strategy of adding curcumin into deprotonated NaCas followed by addition of melted lipid and pectin at pH 12 was more effective in fabrication of uniform and small SLNs (with gastrointestinal-stable), than that of introducing curcumin in ethanol. | [35] |
| Turmeric powder | NLCs | MCT (liquid), GMS (solid); Tween 80 | Hot shear homogenization +ultrasonication | dissolution | 78–93.3%; 40–46.6% | More than 95% of the encapsulated curcuminoids were mainly released during the simulated intestinal digestion; and their bioaccessibility was around 75% (4-fold increase compared to that of free turmeric). | [126] |
| Turmeric extract | NLCs | MCT (liquid), Campritol (solid); poloxamer (emulsifier) | Hot shear homogenization | Dissolution | ~99%; - | The encapsulated turmeric extract in NLCs exhibited good physical stability, higher antioxidant and antimicrobial activities than the free extract. | [125] |

**Table 2.** *Cont.*

| Curcuminoids | Type of Lipid Nanoparticles [a] | Formulation Ingredients (Lipids and Emulsifiers) [b] | Processing Technique [c] | Mode of Drug Incorporation [d] | EE/DL [e] | Main Results | References |
|---|---|---|---|---|---|---|---|
| curcumin | SLNs | Chinese white wax; Tween 20 + lecithin (emulsifiers) | Hot shear homogenization + ultrasonication | dissolution | Max. 84.6%; 10% | Wax SLNs were confirmed to be effective carriers for loading curcumin; the as-fabricated SLNs exhibited a sustained drug release behavior, and better inhibition of the biofilm formation (than free curcumin). | [123] |
| curcumin | SLNs | Tristearin; PEG-modified stearyl ether | Hot shear homogenization +ultrasonication | Mixing (in ethanol) | 91–95%; max 5% | The lipolysis of SLNs was modulated by altering types and concentrations of emulsifiers; high bioaccessibility (>91%) and fast epithelium permeation of encapsulated curcumin in the SLNs were confirmed, resulting in a > 12-fold increase in bioavailability (in rats). | [127] |
| curcumin | NLCs | MCT (liquid), GTS (solid); denatured ovalbumin | Hot shear homogenization +ultrasonication | Dissolution | -; max. 1.0% | It was demonstrated that the oil composition of NLCs produced an influence on transformation, bioaccessibility and intestinal absorption of encapsulated curcumin; NLC containing 20% MCT in the lipid matrix exhibited highest curcumin bioavailability. | [55] |
| curcumin | SLNs | Compritol®888 ATO + GMS; tween 80 + phospholipon 90G | Hot HPH | Mixing (in PEG 600) | ~80%; 15% (*w/v*) | A SLN with high LC (15%) of curcumin was successfully fabricated; the nanoencapsulation enhanced the stability (against photo and chemical degradation) and bioavailability (~70 times higher than free curcumin) of curcumin. | [128] |
| curcumin | SLNs | Propylene glycol monopalmitate, GMS; NaCas-lactose covalent conjugate (emulsifier) | Hot shear homogenization + ultrasonication | Dissolution | >90%; - | The SLNs using NaCas-lactose conjugate as the emulsifier exhibited much better physicochemical properties than those using NaCas alone; the nanoencapsulation improved the antioxidant activity and storage stability of curcumin. | [133] |

**Table 2.** *Cont.*

| Curcuminoids | Type of Lipid Nanoparticles [a] | Formulation Ingredients (Lipids and Emulsifiers) [b] | Processing Technique [c] | Mode of Drug Incorporation [d] | EE/DL [e] | Main Results | References |
|---|---|---|---|---|---|---|---|
| curcumin | SLNs | Compritol 888 ATO; Tween 80 | Modified microemulsion method | Dissolution | 100%; 1.8% | The curcumin in the SLNs exhibited higher antimicrobial effect against *Staphylococcus aureaus* and *Escherichia coli*. | [121] |
| curcumin | SLNs | Murumuru butter; a mixture of Span 20 and Tween 80 | phase-inversion temperature method | dissolution | 98.9%; 1.0% | The curcumin loaded in SLNs was more toxic to colon adenocarcinoma cells. | [132] |

[a] SLNs: solid lipid nanoparticles; NLCs: nanostructured lipid carriers; [b] GTS: glycerol tristearate; MCT: medium chain triglycerides (e.g., Miglyol ®810); GMS: glycerol monostearate; [c] HPH: high pressure homogenization; [d] Curcumin or others were introduced in the lipid nanoparticles, by direct dissolution of powder in melted lipids, or by mixing curcumin dispersion in an organic solvent (or polymer solution) with the melted lipid; [e] EE: encapsulation efficiency (%); DL: drug loading (wt%; relative to total lipid content).

The encapsulation performance of curcumin in SLNs or NLCs is generally determined by its initial solubility in the melted lipid, biocompatibility between the molecule and the lipid matrix (in the solid state), curcumin crystallization, and expulsion from crystallized lipids. In general, NLCs exhibit better encapsulation performance (e.g., higher EE or DL) as delivery nanovehicles for curcumin than SLNs (Table 2). For example, when curcumin was directly added to the melted lipid, NLCs with cetyl palmitate and oleic acid as the solid and liquid lipids, fabricated by a combined hot shear homogenization and ultrasonication, using Tween 80 as the emulsifier, had higher EE of curcumin than SLNs (94% vs. 82%; [134]). Similarly, high EE values (>90%) have been reported for curcumin or turmeric powder-loaded NLCs using MCT as the liquid lipid [52,125,126]. In contrast, SLNs used as nanovehicles for curcumin exhibited a wide range of EE values, from 40% to 100% [39,122–124,127–129,133]. SLNs composed of a sole solid lipid usually exhibited lower EE values than those composed of heterogenous lipids, possibly due to the easier occurrence of lipid crystallization.

For NLCs, the liquid-to-solid lipid ratio is another parameter affecting the encapsulation performance of the system as nanovehicles for curcumin. In curcuminoid-loaded NLCs using MCT as the liquid lipid, Nayak et al. [40] showed that both the EE and DL of curcuminoid in the NLCs progressively increased as the liquid-to-solid lipid ratio increased from 0 to 30%. However, when the ratio was further increased up to 45%, no changes in EE and DL were observed. They attributed the improved encapsulation performance at higher liquid lipid concentrations to increased imperfections within the lipid matrix, as confirmed in the DSC results [40]. The encapsulation performance of curcuminoids in NLCs is also affected by the type and concentration of emulsifiers. In curcuminoids-loaded NLCs, Park et al. [126] observed a slight, but progressive, increase in both EE and DL as the Tween 80 concentration in the aqueous phase increased from 1.0 to 7.5%. Madane and Mahajan [135] also observed a similar increase in EE with increasing Tween 80 concentration in curcumin-loaded NLCs. In contrast, very few works have investigated the maximal loading potential of curcumin in SLNs or NLCs; though, in a few reported works, the DL of curcumin reached 10% or above (relative to total lipid content; [123,124,126,128]. In the case of NLCs as carriers for turmeric powder, DL values of 40–46.6% (relative to total lipid content) were reported [126]. The extremely high DL in these cases could be largely attributable to the high compatibility of the curcuminoid composition in the turmeric powder with the lipid matrix of the NLCs.

The stability of curcumin is expected to be significantly improved by its incorporation into SLNs or NLCs, since the action of oxygen in the solutions is impeded. Noack et al. [52] reported that only around 5% of total curcuminoids encapsulated in the lipid nanoparticles by melt homogenization degraded after 12 months of storage, and the encapsulated curcuminoids did not suffer significant degradation at physiological pHs after 8 hours of storage. Karimi et al. [125] also indicated that the encapsulated curcuminoids in NLCs fabricated by direct incorporation of curcuminoids in the melted lipid exhibited good physical and chemical stability, and only 9% of encapsulated curcuminoids degraded after a storage period of 40 days. Gupta et al. [128] reported that the curcumin encapsulated in SLNs at high DL (15%, *w/v*; relative to the lipid content) did not suffer any significant change in EE and DL upon storage for up to 3 months. In addition to the long-term storage stability, the encapsulated curcumin also exhibited good photo- and pH-stability [128].

When curcumin-loaded lipid nanoparticles can be administered orally or intra-gastrically [127,130], they generally suffer in the digestion and adsorption process. During the gastric and intestinal digestion in the presence of digestive enzymes (e.g., lipases), encapsulated curcumin may be released into the digestive fluids. Park et al. [126] confirmed that, during in vitro simulated digestion experiments, around 95% of the total curcuminoids encapsulated in the NLCs were released during the simulated intestinal digestion, while it less than 2% were released during simulated gastric medium. Ban et al. [127] further indicated that, after the mouth and stomach digestion, the particle sizes of all test SLNs were nearly unchanged (except for the sample suffering serious aggregation during

digestion). Additionally, after full gastrointestinal digestion, the zeta potential of SLNs significantly decreased in magnitude and the particle size increased. The release patterns of curcumin in SLNs and NLCs, as a function of digestion time, during the simulated small intestinal digestion, were consistent with the model used to describe the homogenously-incorporated drug matrix model (Figure 4a) [55,127]. Noack et al. [52] found that the release of loaded curcuminoids in NLCs was much slower than the degradation of the lipid matrix in the simulated intestinal media, and thus considered that the transfer of the curcuminoids into the intestinal media was mainly triggered by the lipid degradation, rather than their release.

To date, a few works have evaluated the absorption of curcumin in the micellar form of undigested lipid nanoparticles (SLNs), using Caco-2 cell monolayers covered with an artificial mucus layer mimicking the human intestinal epithelium [55,122,127]. The curcumin encapsulated in SLN was more readily absorbed by the Caco-2 cells than free curcumin or curcumin in the emulsions [122]. The route of the transport of the SLN across the cells was simple diffusion, and most of the absorbed curcumin (~99%) was metabolized by the cells [122]. Ban et al. [127] further indicated that the transport of curcumin in SLNs through the Coco-2 cell monolayer membrane was passive, and was greatly facilitated by the presence of bile acids and lipases, reflecting that the transfer of encapsulated curcumin in the SLNs to the mixed micelles was favorable for intestinal absorption. Using the same Caco-2 cell absorption model, Feng et al. [55] evaluated the intestinal absorption of micellar curcumin, obtained from NLCs after a simulated gastrointestinal digestion, and showed that the intestinal absorption of micellar curcumin ranged from 26% to 39%, depending on the liquid–lipid (MCT) ratio in total lipid content.

Many in vivo animal experiments indicated that the nanoencapsulation of curcumin in SLNs or NLCs remarkably enhanced the oral or intragastric bioavailability of curcumin (compared to free curcumin suspension), but the extent of improvement considerably varied [38,39,127,136]. Kakkar et al. [130] performed in vivo pharmacokinetics of curcumin in rats after oral administration of curcumin-loaded SLNs, and confirmed a significant improvement in their oral bioavailability (32–155 times) at all doses (1–50 mg/kg dose) as compared to free solubilized curcumin. In a further work, Kakkar et al. [136] showed that, following per oral administration in mice, curcumin-loaded SLNs had a bioavailability 8.1 times greater than that of free curcumin. They also confirmed that curcumin-loaded SLNs could be effectively delivered across the gut wall and blood brain barrier (BBB) and preferentially distributed in the brain of mice (30 times relative to free curcumin). Fang et al. [136] also confirmed that, via intragastric administration in rats, the curcumin-loaded NLCs exhibited a greatly improved ability to cross the BBB, with an 11.9-fold increase in the curcumin concentration in the brain when compared with curcumin suspension. Ban et al. [127] reported that the curcumin encapsulated in SLNs had a more than 12-fold increase in bioavailability compared to the curcumin solution, in a rat model. Gupta et al. [128] developed a novel SLN showing oral bioavailability approximately 70 times higher, with respect to free curcumin, in female Wistar rats. Aside from oral administration, significant improvements of in vivo bioavailability of curcumin using SLNs or NLCs has been demonstrated in animals, e.g., through intravenous administration [137–139].

Antioxidant and anti-inflammatory activities are one of major health benefits of curcumin, as a polyphenol [61,118]. However, these beneficial effects are frequently limited by the water-insolubility and unstable nature of curcumin. Some works indicated that the incorporation of curcumin in SLNs or NLCs did not affect—or slightly but significantly increased—its antioxidant potential, e.g., in vitro DDPH radical scavenging capacity [125,140]. Kakkar et al. [37] showed that oral administration of curcumin-loaded SLNs provided much greater antioxidant protection in vivo (through increased levels of superoxide dismutase, catalase and glutathione, and improved mitochondrial complex enzyme activities) than free-solubilized curcumin, thus resulting in the improvement of cerebral ischemic reperfusion injury in rats. The significantly-improved in vivo anti-inflammatory activity of curcumin via incorporation in SLNs has also been demonstrated

in lipopolysaccharide-induced sepsis of mice, including reduced levels of IL-1β expression and increased anti-inflammatory cytokine IL-10 [141].

The anti-cancer activities of free curcumin in vitro and in vivo have been well recognized [60]. In experiments involving in vitro anti-cancer evaluation against human cancer cell lines (HL-6, A549 and PC3), Vandita and colleagues [142] demonstrated that curcumin loaded in SLNs shared the same anti-cancer mechanisms claimed for free curcumin, including induction of cellular apoptosis by activation of caspases, release of nuclear factor kappa B (NF-κB) activation, and upregulation of TNF-R, but they exhibited significantly higher apoptotic actions against these cancer cells. The curcumin-loaded SLNs and NLCs had improved in vitro antitumor activities in cancer cells (MCF-7 cells, lung cancer cells A549, colon adenocarcinoma cells, prostate cancer cells, or human brain cancer cells A172) [131,132,137,139,143]. The improved in vitro anti-cancer activities of curcumin loaded in SLNs were clearly associated with the enhanced permeability across the cell surface or increased intracellular uptake by the cancer cells. During in vivo animal (mice) experiments with intraperitoneal (i.p.) administration, Chen et al. [137] reported that the incorporation of curcumin in the NLCs enhanced the targeting effect of curcumin to the brain and tumor, and subsequently increased the inhibition efficiency of curcumin from 19.5% to 82.3%, through enhanced apoptosis of cancer cells. They attributed such significant improvements of in vivo anti-cancer activities to enhanced cellular uptake and reactive oxygen species production. Zanotto-Filho and colleagues [144] also showed that curcumin loaded in NLCs exhibited higher cytotoxic activities against gliomas (C6 and U251MG) than free curcumin, through similar mechanisms, including induction of G2/M arrest and autophagy. They confirmed similar anti-cancer activities in rats bearing C6 gliomas in vivo.

In recent years, there has been increasing interest in the neuroprotective effects of curcumin [60]. Kakkar and Kaur [39] evaluated the protective potential of the oral administration of curcumin-loaded SLNs against $AlCl_3$-induced neurotoxicity in mice and, interestingly, found that the oral administration of curcumin-SLNs completely reversed the adverse effects of $AlCl_3$. The loading of curcumin in NLCs was effective at treating amyloid-$\beta$-induced cognitive deficiencies in a rat model of Alzheimer's disease (AD), as a result of a significantly increased accumulation rate of curcumin in the rat brain, as well as its serum levels [140]. The protective role of curcumin-loaded NLCs for brain delivery of curcumin was confirmed by reduced oxidative stress parameters (e.g., ROS formation, lipoid peroxidation, and ADP/ATP ratio) in the hippocampal tissue and improvements in spatial memory [140]. In addition, SLNs have been successfully used as delivery nanovehicles for curcumin to treat major depression disorders [145,146].

4.2.2. Polyphenols

Polyphenols are a kind of phytochemical, found in a wide range of plant-based foods and beverages, including tea, apples, citrus fruit, berries, and coffee. They have been well documented as beneficial for modern human nutrition and health [147,148]. Substantial epidemiological evidence supports close relationships between the administration of polyphenol-rich vegetable and fruits and reduced risks of many chronic diseases, e.g., stroke, myocardial infarction, diabetes, cancers, and hyperglycemia [147,148]. The major polyphenols from dietary plants include flavonols (e.g., quercetin), flavon-3-ols (e.g., catechin and epicatechin), flavanones (e.g., hesperetin), flavones (apigenin and luteolin), isoflavones, anthocyanins, proanthocyanidins, hydroxycinnamates, and phenolic acids [147]. Many polyphenols, e.g., resveratrol, are extremely poorly soluble in water, and susceptible to degradation upon storage or when exposed to lights; they may also undergo extensive first-pass metabolism, resulting in low bioavailability [149]. For example, the oral absorption of resveratrol in humans is high (~75%), primarily through transepithelial diffusion, but its bioavailability is less than 1%, due to extensive metabolism in the intestine and liver [149]. Other works have confirmed that lipid nanoparticles (SLNs and NLCs) can perform as promising and effective delivery carriers for improving water solubility,

stability and bioavailability of such poorly soluble polyphenols, e.g., quercetin [150–154], resveratrol [155], or morin [156].

Similar to curcuminoids, poorly soluble polyphenols can be incorporated into the lipid matrices of SLNs or NLCs through direct solubilization in melted lipids [150–152], or initial solubilization in organic solvents prior to mixing with melted lipids [153–156]. Quercetin is one of the most important and investigated flavonoids to be encapsulated in lipid nanoparticles for improved bioavailability and benefiting effects. In an investigation of the structure–antioxidant activity relationships of several flavonoids (myricetin, quercetin, kaempferol, luteolin, apigenin, and chrysin), quercetin (having four hydroxyl groups) was demonstrated to exhibit the highest activity [157]. Furthermore, it has been documented that quercetin is a more potent antioxidant and radical scavenger than vitamin C, vitamin E, and $\beta$-carotene [158]. In order to improve the therapeutic efficacy of this flavonoid in Alzheimer's disease by utilizing its outstanding antioxidant potential, Dhawan et al. [152] developed a Tween 80-stabilized SLN, fabricated using a microemulsion template method, with quercetin directly incorporated in the melted lipids. It exhibited a high entrapment efficiency of around 86%, and demonstrated that, through intravenous administration, the as-fabricated quercetin-loaded SLNs could effectively cross the BBB, and were successfully target-delivered to the brains of rats. Aditya et al. [150] compared the effectiveness of using SLNs, NLCs, and lipid nanoemulsions (LNE) as delivery nanovehicles for quercetin, indicating that all the test nanovehicles had a high EE of quercetin (>90%) and were stable in simulated stomach conditions. The maximum bioaccessibility was observed with NLC and LNE (~60%), compared to SLN (~35%) and free quercetin solutions (~7%). They also confirmed that both the encapsulated quercetin in the SLN and NLC exhibited a controlled release behavior in simulated intestinal fluid [150]. The much greater bioaccessibility of quercetin in SLN (or NLC), as compared to free quercetin, in fact, was well in agreement with the much greater bioavailability of quercetin encapsulated in SLNs (relative to free quercetin suspension), as confirmed in an earlier work by Li et al. [153]. In situ perfusion experiments in rats indicated that most of the encapsulated quercetin in the SLNs was mainly absorbed in the ileum and colon [153].

Resveratrol is another important polyphenol found in various common foods like grapes, berries, and red wines. It is recognized as a bioactive nutraceutical with a number of potential beneficial effects on health [159]. Teskac and Kristl [160] evaluated the in vitro cellular uptake, transport, and internalization of resveratrol-loaded SLNs in keratinocytes, and found that the SLNs could readily cross the cell membrane in a short period (<15 min), and exhibited greater cytostatic effects, intracellular delivery, solubility, and stability, compared to the free resveratrol suspension. Carlotti and colleagues [161] showed that, in addition to the improved in vitro cell uptake, the resveratrol encapsulated in SLNs had better anti-lipoperioxidative activity. In another work, it was confirmed that the incorporation of resveratrol in the SLNs conferred physical and chemical stability and also provided a sustained release behavior to the encapsulated molecule [57]. Additionally, Pandita et al. [154] further indicated that the encapsulation of resveratrol in SLNs resulted in a significant 8.035-fold improvement in the oral bioavailability of resveratrol (in rats) as compared to the free resveratrol suspension. Jose et al. [155] demonstrated that, through intraperitoneal administration, the encapsulated resveratrol in the SLNs could be preferentially transferred to the brain, with the brain concentration of resveratrol around 5 times higher than that for free resveratrol.

### 4.2.3. Phytosterols and Phytostanols

Plant sterols (or phytosterols) are a family of compounds with structures similar to cholesterol, which have been well shown to lower cholesterol levels when administered orally, irrespective of the present form (esterified or free) [162,163]. In 2000, the FDA allowed a health claim for reducing the risk of coronary heart disease for foods containing plant stanyl and steryl esters, as long as the foods were low in saturated fat and low in cholesterol [164]. Aside from their cholesterol-lowering properties, phytosterols (and

phytostanols) possess anti-cancer, anti-inflammatory, anti-atherogenicity, and anti-oxidant activities [162]. In early efforts to utilize these properties, relatively large dosages of free phytosterols contained in phytosterol-enriched products were required to ensure their cholesterol-lowering benefits [164]. Currently, however, if formulated at optimized conditions, a low dosage (e.g., 1–3 g per day) of phytosterols or their conjugates is enough to exhibit health benefits. Due to this, lipid nanoparticles (SLNs and NLCs) are promising nanovehicles for improving the bioavailability and cholesterol-lowering efficacy of phytosterols, thanks to the good compatibility between them.

To date, only a limited number of works investigated the potential and effectiveness of using lipid nanoparticles as vehicles for improving the stability, release, and bioactivies of phytosterols [165–168]. Such lipid nanoparticles, in this case, are generally NLCs, in agreement with the fact that free phytosterols are poorly soluble in the lipid matrix, and very prone to aggregation and crystallization when crystallization of the phytosterol-containing lipid matrix occurs. Lacatusa et al. [165] fabricated a series of β-sitosterol (a major phytosterol)-loaded NLCs, through a melt emulsification-cooling process, by dissolving it in chloroform prior to the addition of the solution into the melted lipid phase (at 85 °C). In so doing, they confirmed the effectiveness of NLCs, acting as efficient nanovehicles for β-sitosterol, with high EE values of 71–86%. They demonstrated that the β-sitosterol-loaded NLCs exhibited a much greater free oxygen radical scavenging capacity, compared free β-sitosterol (92% vs. 36.5%), and greater β-sitosterol-sustained release behavior, as compared to their counterpart nanoemulsions [165]. In this previous work, it was further indicated that, before the mixing of phytosterol and the melted lipid, the addition of a natural antioxidant, e.g., green tea extract, even at a low concentration, was able to significantly improve the encapsulation performance and antioxidant potential of phytosterols in such nanoparticles.

Although the solubility of phytosterols in common edible triacylglycerols at room temperature is very limited (2–3 wt%), it progressively increases up to 40–50 wt% (in MCT or trilaurin) at temperatures of around 100 °C [166]. According to this solubility-temperature profile, Soleimanian et al. [167] successfully fabricated β-sitosterol-loaded NLCs consisting of propolis wax and pomegranate seed oil (PSO) as the solid and liquid lipids, respectively. They utilized the conventional melt-emulsification (ultrasonication)-cooling process, wherein β-sitosterol, at concentrations up to 15% (relative to total lipid content), was directly formulated in the melted lipid phase. At optimal processing conditions, the EE of β-sitosterol in the NLCs reached 97%, and more importantly, the majority of the crystalline β-sitosterol particles were transformed into the amorphous form [167]. If melt emulsification (to form nanoemulsions) is performed using a microfluidizer (or ultrahigh HP homogenizer), phytosterol-loaded SLNs with an amorphous phytosterol state in the lipid matrix can be formed through a rapid temperature-quenching process, thanks to the formation of finer and more homogenous nanoemulsions [166]. In a more recent work, using the same strategy to fabricate β-sitosterol-loaded NLCs as in the previous work [167], Soleimanian et al. [168] investigated the physical and oxidative stability, in vitro, digestion, and hypocholesterolemic activity, of β-sitosterol loaded in NLCs. They indicated that the as-fabricated NLCs exhibited greater physical stability during long-term storage and during simulated mouth and stomach digestion, as well as higher overall bioaccessibility at the end of digestion. They justified the remarkable affinity of β-sitosterol to the complex lipid mixture formed during the digestion, demonstrating the improved reduction of the encapsulated β-sitosterol (in the NLCs) in the total cholesterol and low-density lipoprotein cholesterol plasma levels in mice, as compared to free β-sitosterol suspension. Lastly, it warrants mention that the formation and stability of free phytosterol-loaded NLCs are also affected by the lipid composition and properties. On that note, da Silva and colleagues [169] investigated the influence of lipid composition and chemical interesterification on the formation and physical stability of free phytosterol-loaded NLCs with soy oil and fully hydrogenated palm oil (HPO) or crambe (HCO) as the liquid and solid lipids, respectively. They indicated that the as-fabricated NLCs with interesterified

blends displayed smaller particle sizes and less particle agglomeration than those with simple blends. The interesterification resulted in a reduction in the melting temperature and recrystallization index of the NLCs. Thus, they demonstrated the good encapsulation efficiency and loading capacity of free phytosterol in NLCs with interesterified blends.

## 5. Strategies to Improve the Performance of SLNs and NLCs as Nanovehicles for Nutraceuticals

Despite the well-recognized consensus that SLNs (and NLCs) exhibit great potential as effective nanovehicles for improving the oral bioavailability of many drugs and nutraceuticals [16,18,56,170], the effectiveness of such nanoparticles is determined by the nature and biocompatibility (with lipid matrix) of nutraceuticals, the stability of global stability during the digestion, the targeted site where the encapsulated bioactive is expected to be released and absorbed, and the in vivo stability of absorbed nutraceuticals in the circulation system. For poorly soluble and highly susceptible nutraceuticals, e.g., curcumin, the loading capacity of such delivery systems is also an important parameter impacting their effectiveness when formulated in foods.

To date, many strategies have been proposed to improve the effectiveness of SLNs (or NLCs) as nanovehicles for nutraceuticals. Such improvements include increased encapsulation performance (e.g., bioactive loading), enhanced oral bioavailability of nutraceuticals, improved stability of nutraceuticals during digestion and circulation, and improved targeted delivery. The bioactive encapsulation performance of such lipid nanoparticles can be improved by selecting an appropriate lipid matrix (solid and/or liquid lipids) in which the to-be-loaded bioactive can be well solubilized, even when the matrix has solidified upon cooling. The composition and nature of the lipid matrix is also important for the stability, release behavior, and bioaccessibility of encapsulated nutraceuticals in such nanoparticles. For example, MCT tends to be digested more rapidly than long-chain triglycerides, but the mixed micelles formed by the digestion products of long-chain triglycerides tend to have higher solubility than those of MCT [24,25]. In this section, we only presented strategies to improve the oral bioavailability of nutraceuticals loaded in such lipid nanoparticles by surface coating or modification techniques, as well as their targeted delivery in vivo.

### 5.1. Improving Oral Bioavailability of Loaded Nutraceuticals in SLNs (or NLCs) by Coating with Chitosan or Its Derivatives

The interfacial structure and surface properties of SLNs and NLCs are of crucial importance for their stability and in vivo oral bioavailability and/or bioactivities. Appropriate surface modification is considered an effective strategy to improve the delivery and bioavailability of drugs or nutraceuticals loaded in the SLNs or NLCs [171]. Surface coating with polymers is frequently used to overcome the defects, e.g., burst release and chemical degradation of encapsulated molecules, occurring in the highly acidic environment of the stomach during oral administration, for SLNs or NLCs, as small intestine-targeted nanovehicles of nutraceuticals. The potential effectiveness of many natural and synthetic polymers to modify the surface properties of such delivery systems has been well investigated, including chitosan or its derivatives, thiomers, polyethylene glycol (PEG), ligands, and cell-penetrating peptides (Figure 8).

Chitosan is a cationic polysaccharide, and as a non-cytotoxic polymer, it is well known for its good mucoadhesive properties and penetration-enhancing properties across mucus epithelia in the digestive tract [172,173]. It can readily adsorb to the negatively-charged surface of common anionic surfactant-stabilized lipid nanoparticles (SLNs or NLCs) through electrostatic interactions [103] and, as a result, a viscoelastic interfacial film coating these nanoparticles is formed, resulting in surface modification. Due to the cationic charge of chitosan, the internal structure stability of chitosan-coated SLNs or NLCs in the gastric environment is enhanced, and the action of lipases on these nanoparticles is, to a certain extent, inhibited. Furthermore, the residence time of lipid nanoparticles in the gastrointestinal tract is also increased, thanks to the adhesive property of chitosan, which leads to enhanced absorption of the nanoparticles by passive diffusion, and ultimately, improved systemic

circulation time and bioavailability of bioactives loaded in these nanoparticles [174]. In an intestinal epithelial cell model (HT-29/B6), it was found that the addition of chitosan (0.005%) induced a fast decrease in transepithelial resistance due to the alteration in intracellular pH caused by the activation of a chloride-bicarbonate exchanger, leading to the opening of tight junction (Tj) proteins [175]. To date, the enhancement of oral intake or bioavailability of bioactives or drugs loaded in SLNs or NLCs by surface coating with chitosan has been reported for curcumin [124], vitamin D [103], silybin [176], docetaxel [177], and cyclosporine A (an extremely hydrophobic cyclic undecapeptide; [178]). The enhancement mechanism was attributed to the enhanced internalization/uptake of the nanoparticles [176,178], enhanced physical stability in gastrointestinal tract (e.g., for curcumin; [124]) and controlled drug release [174].

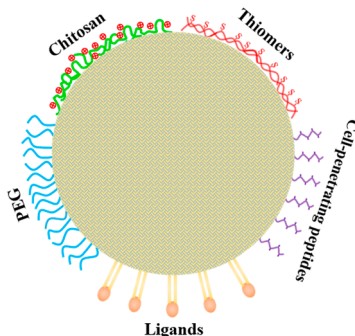

**Figure 8.** Schematic representation of several types of surface-modified SLNs.

Although chitosan coating on the surface of lipid nanoparticles greatly improves the bioavailability of the loaded bioactives, the chitosan-coated lipid nanoparticles are relatively unstable in the intestinal tract, which can possibly lead to the burst release of loaded bioactives and subsequently relatively low bioavailability. This disadvantage has been overcome by using modified chitosan [177,179–181]. Shi et al. [177] synthesized hydroxypropyl trimethyl ammonium chloride chitosan (HACC), which showed pH-independent solubility and electro-positivity. They indicated that the SLNs coated with HACC were highly stable in simulated gastric and intestinal fluids, exhibiting much greater oral bioavailability of the loaded docetaxel (compared to free molecules) in rats. Ramalingam and Ko [181] confirmed that, as compared to unmodified chitosan-coated SLNs, N-trimethyl chitosan-coated SLNs exhibited more prolonged stability and controlled drug release in the simulated intestinal fluid, and significantly higher oral bioavailability and brain distribution of loaded curcumin. In this work, it was also confirmed that the burst release of curcumin in the gastric environment was effectively prevented [181]. Perteghella et al. [180] reported that coating with *N*, *O*-carboxymethyl chitosan (NCC) not only protected oxidative and hydrolytic degradation of curcumin loaded in the SLNs upon long-term storage, but also enhanced cytocompatibility with Caco-2 cells. In fact, in an earlier work, it was indicated that the NCC-coated SLNs exhibited suppressed burst release in simulated gastric fluid, but a sustained release in simulated intestinal fluid; NCC-SLN also exhibited enhanced cytotoxicity and cellular uptake on breast cancer cells (MCF-7 cells) [179]. As expected, the lymphatic uptake and oral bioavailability of NCC-SLN were much higher than free curcumin [179].

### 5.2. Improving the Transport across Intestinal Barrier and Target-Delivery of Loaded Nutraceuticals by Surface Modifications with Functional Molecules

In addition to the enhanced oral bioavailability, surface modification of lipid nanoparticles (SLNs and NLCs) with functional molecules can increase the stability of nutraceuticals loaded in such nanoparticles in systematic circulation [182], transport of drugs across intestinal barriers [183], and their targeted delivery [184].

In addition to the positively charged chitosan, many thiolated polymers with sulfhydryl groups, often termed as thiomers, have been developed to modify the mucoadhesive properties of lipid nanoparticles [184]. Thiomers can form disulfide bonds between sulfhydryl

groups on thiomers and cysteine, rich on glycoproteins, in the mucus layer of the gastrointestinal tract, thus improving the adhesion properties, prolonging the retention time in the gastrointestinal tract, and enhancing the penetration and absorption of drugs [185,186]. Furthermore, due to the interaction between thiol groups and the transmembrane region of the efflux pump, thiomers can inhibit P-glycoprotein activity in small intestinal epithelial cells, thereby enhancing drug absorption efficiency [187]. Cysteine and its derivatives are common modifiers of thiolation. Fang et al. [185] reported that the surface modification of docetaxel-loaded NLCs with cysteine significantly improved their adhesion properties and absorption efficiency, and the oral bioavailability levels of docetaxel loaded in the cysteine-functionalized NLCs were 12.3-fold and 1.64-fold higher than free docetaxel and unfunctionalized NLCs, respectively. Tian and colleagues [188] used N-acetyl-L-cysteine to modify curcumin-loaded NLCs, and showed that the oral bioavailability of curcumin in rats was positively correlated with the degree of surface modification of NLCs, and the bioavailability of curcumin in the optimal group was 499.45-fold and 116.89-fold higher than free curcumin and curcumin loaded in unmodified NLCs, respectively.

Cell-penetrating peptides are a class of small molecule peptides with transmembrane transport ability, generally consisting of 5–30 amino acid residues. They are rich in basic amino acids, polycationic and amphipathic [189]. These peptides are electrostatically interacted with, or covalently bound to, a variety of exogenous substances, such as peptides, oligonucleotides, liposomes, or SLNs. It is generally recognized that these peptide complexes or conjugates enter cells by two mechanisms of membrane penetration: non-energy-dependent direct penetration and energy-dependent endocytosis, though the mechanism of membrane penetration remains controversial [190]. Fan et al. [183] prepared two cell-penetrating peptide-modified SLNs loaded with salmon calcitonin, which had a greater ability to promote cellular internalization and mucosal penetration than unmodified SLNs. The mechanism of action was mainly active transport via latticin and fossa-dependent endocytosis. Cell-penetrating peptide modification of SLNs enhanced the oral bioavailability of the active ingredient [183]. Meng et al. [184] reported that the surface modification of NLCs with lactoferrin produced novel low-density lipoprotein (LDL)-mimic NLCs which could perform as brain-targeted delivery nanocarriers for curcumin. They demonstrated that the modification greatly improved the uptake of surface-modified NLCs in the brain capillary endothelial cells; the as-fabricated LNCs effectively permeated brain blood barriers (BBB) and preferentially accumulated in the brain (2.78 times greater than those loaded in the unmodified NLC) [184].

A variety of specific receptors exist on cells, and the transport of active molecules in lipid nanoparticles can be enhanced by surface modification with specific ligands. Considering that bile acid transporters are available on intestinal epithelial cells, Tian and colleagues [191] demonstrated that the surface modification of NLCs with taurocholic acid remarkably improved the intestinal absorption and oral bioavailability of curcumin loaded in the NLCs. The AUC of curcumin, after oral administration in rats, was 5–15-fold higher than that of unmodified NLCs at different levels of modification, suggesting that the addition of taurocholic acid ligands to NLCs promoted curcumin absorption [191]. The folate receptor is a ubiquitous receptor on cells, but it is overexpressed in a variety of solid tumor cells. The use of folic acid for targeted delivery of antitumor drugs has been heavily investigated for the ability of folic acid-bound molecules or delivery vehicles to enter cells via folic acid receptor-mediated endocytosis [192]. Folic acid-decorated NLCs were constructed for the delivery of curcumin nanodrugs, which effectively increased the inhibition of curcumin against MCF-7 human breast cancer cells and showed better in vivo antitumor activity [193]. In another study, it was indicated that the folic acid functionalization of docetaxel- and curcumin-coloaded SLNs greatly increased their cellular uptake and cytotoxicity, and the simultaneous co-encapsulation enhanced the synergistic activity of these two drugs [194].

## 6. Concluding Remarks and Future Research Prospects

Over the past few decades, it has been widely recognized that the development of nanosized delivery systems is one of the most effective strategies to improve the stability and oral bioavailability of many poorly soluble nutraceuticals. Among all these nano-delivery systems, SLNs and NLCs seem to be the most appropriate delivery systems for lipophilic or hydrophobic nutraceuticals, thanks to their good compatibility with lipid matrices. The usage of such nanoparticles as nanovehicles for nutraceuticals exhibits many advantages over other nano-delivery systems: for example, high loading capacity, good biocompatibility, high stability (especially oxidative stability), prolonged release, high cellular uptake and bioavailability, and easy scale-up. In contrast, NLCs as nanovehicles for nutraceuticals or bioactives perform better, in terms of loading capacity and inhibition of perfect lipid crystallization, than SLNs.

Although SLNs and NLCs have been successfully developed as nano-delivery systems for drugs for several decades, the use of such nanoparticles as food-grade nanovehicles for nutraceuticals has only come to pass within the past two. To date, such nanoparticles have been confirmed to act as effective nanovehicles for liposoluble nutraceuticals, e.g., carotenoids, fat-soluble vitamins, $\omega$-3 polyunsaturated fatty acids, and essential oils, as well as for poorly soluble bioactive compounds, e.g., curcuminoids, polyphenols, phytosterols, and phytostanols. Accumulating evidence suggests that SLNs and NLCs exhibit great potential as nanovehicles for functional food formulations with beneficial effects on the brain and human health. In this regard, one successful example is the development of curcumin-loaded SLNs or NLCs for the prevention and improvement of many common central nervous system diseases, including Alzheimer's disease, Parkinson's disease, ischemic stroke, and multiple sclerosis [17,62]. This is closely associated with the fact that lipid nanoparticles (including SLNs and NLCs) can be directly absorbed by the epithelium cells in the intestinal tract and enter the lymphatic system.

Despite their promising potential as nanovehicles for lipophilic or hydrophobic nutraceuticals, great effort is still needed for the development and utilization of food-biocompatible SLNs and NLCs as effective nanovehicles that will significantly improve their oral bioavailability. Future research prospects for the development of such nanoparticles as food-grade nanovehicles for nutraceuticals should include, but not be limited to, the following aspects:

(i). elucidating the potential and effectiveness of SLNs and/or NLCs as nanovehicles for nutraceuticals with specific properties, e.g., heat-labile or lipid-insoluble bioactives;

(ii). characterizing the stability and absorption of SLNs and/or NLCs during in vitro simulated digestion and unravelling the relationships between the bioaccessibility and bioavailability of encapsulated nutraceuticals;

(iii). developing surfactant-free, food-grade SLNs (or NLCs) as oral nanovehicles for the delivery of nutraceuticals or bioactives;

(iv). elucidating the health and function-related effectiveness of encapsulated nutraceuticals, e.g., using in vitro cell models or in vivo animal models;

(v). determining the suitability of incorporating SLNs and NLCs into different types of food formulations, e.g., beverages and milks, and investigating effective drying techniques for obtaining powdered products.

**Author Contributions:** Conceptualization, C.-H.T.; methodology, C.-H.T.; software, J.-R.D.; validation, C.-H.T., H.-L.C. and J.-R.D.; resources, H.-L.C. and J.-R.D.; data curation, J.-R.D.; writing—original draft preparation, H.-L.C.; writing—review and editing, C.-H.T.; visualization, C.-H.T.; supervision, C.-H.T.; project administration, C.-H.T.; funding acquisition, C.-H.T. All authors have read and agreed to the published version of the manuscript.

**Funding:** This work was funded by the National Natural Science Foundation of China, grant numbers 32172343 and 21872057.

**Informed Consent Statement:** Not applicable.

**Data Availability Statement:** Not applicable.

**Conflicts of Interest:** The authors declare no conflict of interest.

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
