# Peer review of "Solid Lipid Nanoparticles (SLNs) and Nanostructured Lipid Carriers (NLCs) as Food-Grade Nanovehicles for Hydrophobic Nutraceuticals or Bioactives"

_applsci, doi:10.3390/app13031726_

Round 1

Reviewer 1 Report

Chuan-He Tang et al write an excellent review on Solid lipid nanoparticles (SLNs) and nanostructured lipid carri- 2 ers (NLCs) as food-grade nanovehicles for hydrophobic 3 nutraceuticals or bioactives. This review article is informative and well balanced. I recommend for publication in present form.

Author Response

Thanks for the very highly appreciated comments of the reviewer. 

Reviewer 2 Report

1.       This review article comprehensively covered the three major areas: Preparation method, key issues, and recent advanced development, and improvement of the oral bioavailability.

2.       Great reference for both academia and industry in terms of the major factors need to be taken into consideration such as emulsifiers, co-emulsifiers, as well as very detailed methods of how to make SLNs.

3.       This review is very comprehensive and well-organized, especially the summary of some representative works in the table.

4.       Section 5 summarized the strategy to improve the performance of SLNs and NLCs nanovesicles, which provided many valuable references for future studies.

5.       What are the main challenges (economic or technical) in terms of the massive production of these SLNs and NLCs or? Is there any advanced methods or process could facilitate the development of SLNs in the industry?

6.       It looks like there are two different fonts in Figure 5. The “Lipid nanoparticles (e.g., SLNs or NLCs) look like a different font. Please fix it.

7.       The resolution of the chemical structure in Figure 7. Is poor. Please consider using Chemdraw or other method to fix it.

Author Response

We are very grateful for the generally positive comments about our manuscript. The responses to the comments of the reviewer (point-to-point) are lised as follows:

  1) This review article comprehensively covered the three major areas: Preparation method, key issues, and recent advanced development, and improvement of the oral bioavailability.

     Reply: Thank for the careful review about our manuscript.

2) Great reference for both academia and industry in terms of the major factors need to be taken into consideration such as emulsifiers, co-emulsifiers, as well as very detailed methods of how to make SLNs.

    Reply: Although we agree it would be better to take many references (suggested by the reviewer) into consideration, it is a really time-consuming work. 

3) This review is very comprehensive and well-organized, especially the summary of some representative works in the table.

    Reply: Thank you for your very impressive comments.

4) Section 5 summarized the strategy to improve the performance of SLNs and NLCs nanovesicles, which provided many valuable references for future studies.

    Reply: Thank you.

5) What are the main challenges (economic or technical) in terms of the massive production of these SLNs and NLCs or? Is there any advanced methods or process could facilitate the development of SLNs in the industry?

     Reply: According to our recent experiences in the field, we think the main challenge for the massive productive of these lipid nanoparticles is how to efficiently prepare the nanoemulsions, prior to the formation of lipid nanoparticles. In general, a mass concentration of surfactants is required to fully cover the interface of the newly formed droplets, during emulsification which is usually performed at high temperatures (ensuring that all the lipid in the emulsions is in the liquid state). 

At high temperatures, surfactants initially adsorbed at the interface are easily desorbed. Furthermore, the addition of high concentrations of surfactants may cause a safety issue. Due to this, in the food field, many food-grade emulsifiers (e.g., proteins) are used instead of the small surfactants.

In our recent works, we have successfully utilized the proteins as the emulsifiers to form lipid nanoparticles, and obtained a patented technology.

6) It looks like there are two different fonts in Figure 5. The “Lipid nanoparticles (e.g., SLNs or NLCs) look like a different font. Please fix it.

   Reply: Yes. Lipid nanoparticles have two different dimensions of size. 

7)   The resolution of the chemical structure in Figure 7. Is poor. Please consider using Chemdraw or other method to fix it.

    Reply: We have enlarged the Figure 7, to make them more visible. Thanks.

Reviewer 3 Report

Congratulation for your work. I don't have any negative comment to send to you

Author Response

We are very grateful for the highly positive comments about our manuscript.

Reviewer 4 Report

I think that the topic of this review, namely “Solid lipid nanoparticles (SLNs) and nanostructured lipid carriers (NLCs) as food-grade nanovehicles for hydrophobic nutraceuticals or bioactives” is really interesting and suitable for the «Applied Sciences» Journal. This review is comprehensive, clear and well written. It covers both basic and innovative works of recent years. The description of the considered approaches can be useful for both beginners and experts in the field of lipid-based nanocarriers for nutraceuticals. Only some clarifications are still required:

Line 88: Does this mean not LA, but LC or something else?

Line 105: Please explain here what is the difference between hydrophobic and lipophilic nutraceuticals?

Line 156: Do you mean stearic acid, not steric acid?

Lines 241-244: Please specify which lipid was taken as the basis for the SLN.

Line 245-246: Please indicate in which part of the gastrointestinal tract the SLNs showed an exceptional stability and controlled release of curcumin?

Figure 3b: Please clarify, in case "b" microemulsions are obtained, and not nanoemulsions?

Lines 360-363: This is a repeated statement.

Lines 534-549: These sentences are already given verbatim in the text, so there is no need to repeat them in the figure caption!

Lines 642-643: Perhaps the title of this section could be changed here, since the entire manuscript is a review.

Line 712: Maybe the authors meant esters, not asters?

Lines 1044-1045: (Siviero, Gallo,Maggini, Gori, Mugelli, Firenzuoli, & Vannacci, 2015) - this is the reference -120.

Line 1058: There seems to be a mistake in the name of bisdemethoxycurcumin.

Line 1497: (Meng et al., 2015)- this is the reference -184.

Author Response

I think that the topic of this review, namely “Solid lipid nanoparticles (SLNs) and nanostructured lipid carriers (NLCs) as food-grade nanovehicles for hydrophobic nutraceuticals or bioactives” is really interesting and suitable for the «Applied Sciences» Journal. This review is comprehensive, clear and well written. It covers both basic and innovative works of recent years. The description of the considered approaches can be useful for both beginners and experts in the field of lipid-based nanocarriers for nutraceuticals. Only some clarifications are still required:

  Reply: We are very appreciated for the highly valued comments about our manuscript by the reviewer.

Line 88: Does this mean not LA, but LC or something else?

   Reply: Here, it had been corrected as 'LC'. Thanks.

Line 105: Please explain here what is the difference between hydrophobic and lipophilic nutraceuticals?

   Reply: Here, we have made an explanation about this, as follows: 'hydrophobic (poorly water-soluble) or lipophilic (oil-soluble)'.

Line 156: Do you mean stearic acid, not steric acid?

   Reply: Corrected. Thanks.

Lines 241-244: Please specify which lipid was taken as the basis for the SLN.

    Reply: All the works performed by the Prof. Luo' group used Compritol ATO 888 (glyceryl behenate) as the lipid phase. We have specified this in the section. Thanks.

Line 245-246: Please indicate in which part of the gastrointestinal tract the SLNs showed an exceptional stability and controlled release of curcumin?

      Reply: It should be in the stomach part wherein the SLNs exhibited an exceptional stability and controlled release of curcumin.

Figure 3b: Please clarify, in case "b" microemulsions are obtained, and not nanoemulsions?

  Reply: Corrected.

Lines 360-363: This is a repeated statement.

   Reply:  We removed this repeated statement.

Lines 534-549: These sentences are already given verbatim in the text, so there is no need to repeat them in the figure caption!

   Reply:  We agree with this. In the revised version of manuscript, we have removed the statements in the figure caption. Thanks.

Lines 642-643: Perhaps the title of this section could be changed here, since the entire manuscript is a review.

    Reply: We have made accordingly a revision to make it more appropriate.

Line 712: Maybe the authors meant esters, not asters?

    Reply: Corrected.

Lines 1044-1045: (Siviero, Gallo,Maggini, Gori, Mugelli, Firenzuoli, & Vannacci, 2015) - this is the reference -120.

    Reply: We have corrected this typo error. Thanks.

Line 1058: There seems to be a mistake in the name of bisdemethoxycurcumin.

    Reply: We have checked this. It is correct. 

Line 1497: (Meng et al., 2015)- this is the reference -184.

    Reply:  Thank you for your very responsible review.